# Examining the Interplay between CEPSA's ESG Performance and Financial Performance: An Overview of the Energy Sector Transformation

**Yangxueyi Hu** [1], **Abeer Hassan** [2] and **Sehrish Atif** [2,*]

1   Adam Smith Business School, University of Glasgow, Glasgow G12 8QQ, UK; huyangxueyi@outlook.com
2   School of Business & Creative Industries, University of the West of Scotland, Paisley PA1 2BE, UK; abeer.hassan@uws.ac.uk
*   Correspondence: sehrish.atif@uws.ac.uk

**Abstract:** This study delves into the financial performance of the Compañía Española de Petróleos, S.A.U. (CEPSA) within the context of the ongoing ESG transformation in the Energy Sector. The primary aim of this research is to understand the critical dimensions essential for evaluating energy companies' ESG performances. The research assesses the changes in CEPSA's financial indicators over the last five years (2018–2022). The report uses DuPont analysis to evaluate CEPSA's environmental and social responsibility performances. The study examines several financial performance metrics, including return on net assets, profitability, and corporate financing structure changes. The methodology of this study comprehensively assesses CEPSA's sustainable development trajectory and ESG management system. The analysis reveals that CEPSA has consistently improved its sustainable development capabilities over the last five years by establishing a comprehensive ESG management system. While return on net assets and profitability indicators have shown positive trends, the financing structure has changed significantly. Notably, the proportion of debt financing has increased substantially, and there is a slight decline in the net profit margin. The formal transformation in 2020 further influenced increases in liabilities and fixed assets for CEPSA. The study focuses on CEPSA's sustained improvements in ESG management and the associated shifts in financial metrics, adding originality to the study and offering a nuanced perspective on the evolving landscape of sustainable practices. The study reveals the financial implications of ESG transformation in the energy sector and offers valuable insights for stakeholders. Moreover, this research contributes to the existing literature by employing the DuPont analysis system to explore the intricate relationship between ESG performance and financial indicators in the energy sector.

**Keywords:** ESG performance; financial performance; energy transformation

## 1. Introduction

According to the United Nations Sustainable Stock Exchange (SSE) initiative, all large companies are required to publish reports on their environmental and social practices by 2030 [1]. The depletion of natural resources, increased global warming, air pollution, soaring energy prices, and epidemics resulting in crude oil price volatility have led to an awareness of the importance of ESG transformation and enhanced energy management [2]. These phenomena appear to be only social and environmental problems; however, their negative economic impact cannot be ignored, so it is imperative to reduce carbon emissions, lower the carbon intensity of enterprises, increase the ratios of renewable energy use and energy utilisation, and achieve sustainable development [3]. As one of the main GHG emission sectors, the energy sector's successful transition is key to global decarbonisation. This is a challenge for the energy sector, which always relies on natural resources. If the energy sector can be successfully transformed, it will bring a new dynamic to the world economy.

However, some unavoidable issues exist when assessing the ESG performance in the energy sector. First, ESG performance analysis lacks a specific analytical framework. Most researchers rely on ESG scores published by rating agencies as the basis for measuring corporate ESG performances [4]. However, rating agencies seldom announce the factors affecting ESG scores in each dimension, so it is difficult for enterprises to clarify the path of transformation without knowing the reasons for score changes, and it is further difficult to predict the challenges and opportunities in the way of transformation. This is also unfavourable for enterprises to establish internal evaluation systems and control risks in advance and obtain timely feedback.

Second, the topic lacks a case study. Large-sample trend analysis focuses on numerical changes between ESG scores and financial performance, but it lacks an exploration of the internal causes that lead to changes in ESG performance. Meanwhile, the ESG scores of rating agencies do not represent companies' actual ESG performances. The scores of rating agencies are based on the enterprises' ESG disclosure level, but enterprises with a weak sustainability ability are suspected of "greenwashing" themselves by disclosing more ESG information [4]. Therefore, it is more important to analyse an enterprise's strategic planning, financial capital status, and actual business activities rather than a single score.

Finally, the energy industry is under great pressure to transform and has a significant impact on upstream and downstream customers. Unlike the financial, healthcare, and food industries, the energy sector, especially oil and gas processing companies, is often environmentally destructive and has a high carbon intensity, making the path to transformation more difficult. At the same time, the products of the energy sector are the basis of many industries, such as fuel oil produced by refineries, electricity produced by generators, and sub-products processed from petroleum. A successful transition in the energy sector can help downstream customers decarbonise and enhance the sustainability of a wider range of industries. However, energy decarbonisation cannot be achieved simply by improving technology and adapting business strategies. It also requires enhanced supplier management and the promotion of the ESG transformation of upstream suppliers [5].

As a multinational oil and gas company, the Compañía Española de Petróleos, S.A.U., commonly known as CEPSA, was founded in Spain in 1929, and its main activities include the exploration and production of oil and gas, the research and development of petrochemicals, and the production of green hydrogen and bioenergy. In 2020, CEPSA established a new Environmental, Social and Governance (ESG) Committee and received an A- rating from the Carbon Disclosure Program's Climate Change Initiative. CEPSA's ESG ratings in 2022 were in the range of those of Sustainalytics, S&P Global, Moody's, clarity AI, and others at the forefront of the oil and gas industry. As a diversified energy company, CEPSA's vision is to lead the way in sustainable energy and mobility, and to achieve this, they launched the 2030 Positive Motion strategy in 2022 [6]. This strategy is committed to supporting customers on their decarbonisation journey, bringing value to society and stakeholders, and achieving business delivery in the exploration and production and chemicals segment [7], which played an important role in CEPSA's ESG development. The case section will summarise the specific factors that need to be considered when analysing the environmental and social responsibility performances of energy companies and will also analyse CEPSA's financial performance after the ESG transition. This paper provides an analytical direction for investors and managers to assess the sustainability of energy companies.

The rest of the paper consists of the following. Section 2 is a literature review, Section 3 explains the data sources and the analytical methodology, followed by a discussion on the findings in Section 4. The last section summarises the paper.

## 2. Literature Review

### 2.1. Related Research on the ESG Performance Evaluation

The ESG performance is a comprehensive evaluation of a company's environmental, social responsibility, and governance capabilities, so it is important to understand the

factors that affect ESG performance before evaluating a company's ESG performance [4]. Drempetic et al. (2020) [4] claimed that a positive relationship exists between firm size, the richness of ESG data resources, and ESG performance. In analysing Thomson Reuters ASSET4 ESG ratings, it was found that large companies disclose more ESG data and more available data to boost ESG scores. However, company size does not enhance a company's sustainability. Taking air pollution as an example, the indicator that measures the level of air pollution is greenhouse gas emissions (GHG emissions); large-scale companies do not have a lower intensity of GHG emissions than small-scale companies, but they present a higher ESG score. When comparing ESG performances across firms cross-sectionally, Garcia et al. (2017) [6] found that industry type is an important influencing factor. Even controlling for objective factors such as firm size and country of listing, companies in sensitive industries typically perform better in terms of environmental performance, as they seek to protect their reputation by disclosing more ESG data. At the same time, there is an inverted U-curve relationship between systemic risks arising from climate change (environmental), economic inequality and working conditions (social), and a company's ESG performance. Theoretically, an optimal ESG performance can be achieved if firms can reasonably control systemic risks [6]. However, an optimal ESG performance is difficult to achieve in actual operations. Galbreath (2013) [7], after observing the ESG performances of 300 Australian listed firms, found that the corporate strategy system may be a driver to improve the ESG performance. After assessing the three components of ESG separately, he found that corporate governance performance improved significantly faster than environmental or social responsibility performances, and therefore, managers were more inclined to improve the ESG performance through corporate governance. Also, the speed of improvement in a particular aspect of corporate ESG performance is influenced by strategy, culture, and risk propensity [8]; for example, companies that focus on humanistic organisational culture tend to perform better in terms of social responsibility.

When evaluating the ESG performance of a specific industry or company, there are often different evaluation methods available, but nowadays, the common evaluation method is based on ESG scores provided by third-party rating agencies. There are two main different approaches. One is to build the researcher's comprehensive social responsibility index [8] based on the KLD (Kinder, Lyndenberg, and Domini) scores in 13 categories. The KLD scores do not provide a comprehensive ESG score for a particular company or provide separate scores for environmental, social responsibility, and corporate governance performances. Instead, it divides ESG performance into seven major categories and six minor categories and scores each company's strengths and failings in each of the 13 categories. Most researchers discard certain categories based on the nature of the industry, e.g., Lins et al. (2017) [9] designed an evaluation system that excludes product and corporate governance elements. Another one is based on the rating agency's overall ESG score for the company or the scores for each component of environmental, social responsibility, and corporate governance. For example, Kim et al. (2013) [9] chose the data provided by MSCI and found that a firm's ESG performance was positively correlated with its stock returns and Tobin's Q in the Korean market over the same period. The change in ESG performance in this research model was based on the ESG rating provided by MSCI, and the researchers have observed the changes in the ESG performance of over 100 companies since 2011.

### 2.2. The Relationship between ESG Performance and Financial Performance

Although the relationship between ESG performance and financial performance has been noted by many scholars, there is still no uniform conclusion [4,10]. Early scholars argued that if a company takes on 'social responsibility', it will inevitably incur additional costs and ultimately affect corporate profits [2,11], which are contrary to the company's best interests. If a company wants to achieve social responsibility without affecting its interests, it will inevitably affect the interests of its stakeholders. Aupperle et al. (1985) [12] directly stated that there is a negative correlation between socially responsible performance and financial performance. This is because the cost of operations increases when a company

fulfils its social responsibility, and the company must take on additional costs that could be covered by the government or individuals. For example, if a company raises its costs by purchasing additional water purification equipment, but a competitor does not have such a plan, the company's profits will be reduced, ultimately affecting shareholder wealth.

However, as the topic has been studied, some scholars have found a positive correlation between ESG performance and financial performance [13], and the effect is bidirectional. Based on different ESG performance evaluation methods, good managers will improve their ESG scores by supporting ESG performance objectives. This helps companies to invest accurately while being below the negative impact of serious risks. Lins (2017) [9] found, after controlling for firm characteristics and risk factors, that during a crisis, firms with high ESG ratings are more able to resist financial risks and have a better financial performance than firms with low ESG ratings. This suggests that the social capital generated by ESG activities is useful when a firm experiences a crisis of confidence. This phenomenon is even more pronounced in the banking sector, where some banks with a good social responsibility performance show a significant positive correlation between ROE and ESG scores [12]. After the 2009 financial crisis, more large banks actively participated in activities to improve their CSR, which also contributed to reducing the likelihood of a recurrence of the crisis [10].

A small number of scholars argue that the relationship between SG performance and financial performance is neutral. McWilliams and Siegel (2001) [14], after conducting analyses using supply and demand theory and cost–benefit theory, claimed that there is no significant change in corporate profits before and after ESG investment. However, their study took profit as an important indicator of a firm's financial performance, and the evaluation criteria were too independent. Meanwhile, the ESG attributes of a company are like other services and goods provided by the company, which need to be assessed, to maximise profits and shareholder wealth. Arlow and Gannon (1982) [15] suggested that for profit-making businesses, sustainability-related attributes should be subordinated to operational attributes. In addition, it is difficult to conclude whether there is a positive or negative relationship between changes in ESG performance and financial performance as business responses following ESG investment are not the same influenced by factors such as the nature of the industry and the size of the organisation [16,17].

Research on the relationship between ESG performance and financial performance has mostly focused on a single dimension. Han et al. (2016) [18] used the scores provided by Bloomberg for three components of ESG in Korean companies, namely the Environmental Disclosure Performance Score (EDS), the Social Disclosure Performance Score (SDS), and the Governance Performance Score (GDS), to analyse the relationship between the changes in scores of these three dimensions and financial performance (FP). Barnett and Salomon (2012) [18] calculated a comprehensive social performance score for 1214 companies based on the 13 component scores provided through KLD scores and then used this to analyse the relationship between changes in the ESG performance and financial performance. Ultimately, they both concluded that there is an anti-U relationship between the ESG performance and FP, implying that investment in ESG activities can improve the financial performance only after a certain accumulation level.

### 2.3. Transforming the Energy Sector for Sustainable Development

Although the relationship between overall ESG performance and financial performance is not yet clear, investing in ESG to enhance a company's reputation and strengthen stakeholder interaction is well established [19]. As a result, more companies published ESG-related policies, and the number of companies voluntarily disclosing ESG information is increasing. Except for the policy guidance, investor attitudes are also a key influence factor for company investment in ESG. Investors rank the scores of different companies based on the ESG dimensions and select public companies with better ESG activity quality in a particular dimension [20]. The financial performance of investee companies in this investment model is improved more significantly, especially for companies that perform well in the EDS. Investors are also more likely to achieve superior returns by taking advantage

of this feature. Meanwhile, an increasing number of stock exchanges around the world are imposing requirements on the level of disclosure of ESG parameters by listed companies. By 2020, 24% of global exchanges will have made this requirement [21]. As more scholars articulate the benefits of incorporating sustainability indicators into credit risk management, creditors are also becoming concerned with companies' ESG performances. Companies with a good ESG performance have lower debt financing costs; conversely, companies with more environmental problems and weak sustainability have higher debt financing costs [22]. Such companies are expected to face higher legal, reputational, and regulatory risks; therefore, their solvency is questioned. In the absence of collateral, creditors are more inclined to choose firms with good ESG performances [3,23].

If the transition to net-zero carbon emissions through economic transformation is to be realised, good policy guidance for the transformation of the energy sector is required [5]. Improvements in energy efficiency are also important for decarbonisation; for example, the power generation industry should aim to provide clean electricity with zero emissions instead of generating electricity from fossil fuel derivatives [24]. Therefore, technological innovation in the energy sector cannot be ignored, and the share of electricity use in final energy use has been increasing in most countries [25], which provides an opportunity to reduce carbon monoxide emissions. In short, the energy sector should seek to decouple CO emissions from the growing electricity demand through technological updates and appropriate emission reduction measures. The decreasing cost of renewable energy generation and the increasing availability of storage technologies have made traditional energy generation less competitive [26]. All these changes are a sign that the traditional energy sector will have to be transformed.

Following the outbreak of COVID-19, economic and tourism activities in various countries were significantly reduced, international trade was substantially affected, and the electricity demand dropped sharply. As a result, the global oil demand fell to its lowest level since 1995 [27]. The epidemic seriously affected the stability of oil and gas prices and undermined the dominance of fossil fuels in the energy sector. Hallegatte et al. (2012) [28] found that growth in the new energy sector will reduce businesses' reliance on imported fossil fuels for production, thereby reducing the impact of energy price volatility. Meanwhile, capital is beginning to flow out of the fossil fuel market as international oil companies significantly reduce their investment programs [29]. On the one hand, these outflows have created space for investment in renewable energy. On the other hand, low oil prices have reduced the return on investment in fossil fuels. To hedge against losses in fossil fuel investments, investors may expand their commercial investments in renewable energy [30]. Therefore, the energy transition is imperative for energy companies, both from the perspective of investment demand and from the perspective of stability of returns [31].

### 2.4. Research Gap

An overview of the existing literature shows that most scholars tend to use correlation and regression to analyse ESG performance and financial performance or use Tobin's Q theory to explain the inter-two variation [5,27]. The selection of research samples is mostly limited to a large sample of one industry or one country. This research approach is beneficial for scholars to identify overall trends but lacks detailed analysis. Firstly, the difficulties faced, and transformation strategies may be different for different industries when transforming their ESG. Even in the same industry, there are differences in the basis of sustainability among companies before transformation due to different business philosophies and product structures [2]. Secondly, most scholars directly refer to the scores of third-party rating agencies when evaluating corporate ESG performances, but at this stage, the scoring methods of rating agencies are not uniform, and some agencies' evaluation systems are not comprehensive [32]; therefore, the comparability between scores is weak. Finally, using ESG rating scores to represent ESG performance has limitations; ESG scores may not be representative of a company's actual ESG activities. In other words,

a company's ESG disclosure activity index may not be consistent with the type of ESG efforts they make.

Based on the above problems, this paper poses the following three research questions:

RQ1. What are the main indicators to focus on when evaluating the ESG performance (environmental and social performances) of energy companies?

RQ2. What are the main ESG activities that an energy company should undertake if it wants to transform itself while controlling costs?

RQ3. How does the financial performance of an energy company change after implementing a sustainable development strategy?

## 3. Methods

### 3.1. Legitimacy Theory

Information is important for evaluating the sustainability of an organisation, but why should an organisation use its limited resources to provide information? Organisational legitimacy and profit orientation are common approaches used to explain this issue. Scholars who support profit orientation believe that an ESG performance improvement contributes to financial performance improvement, so firms disclose more ESG information to improve sustainability to obtain a good reputation and improve profit [9]. At the same time, to reduce the impact of negative ESG information on financial performance, some companies have presented the phenomenon of "greenwashing". They invest money to create an environmentally friendly image for themselves, but do not invest in green projects [33,34]. Scholars who support organisational legitimacy argue that firms enhance sustainability to ensure the legitimacy of business activities [35], with the main theories including new institutional theory and stakeholder theory. Stakeholder theory, which is central to the ESG theme, emphasises that a company's ability to create sustainable wealth depends on its relationships with various stakeholders [36,37]. Therefore, companies should disclose as much ESG information as possible to reduce information asymmetry for the public and increase investor confidence [38,39]. However, the trade-off theory suggests that too much ESG input will increase firms' costs and affect the efficiency of resource utilisation, which will inevitably affect shareholders' wealth [40]. Therefore, this paper hopes to use stakeholder theory and trade-off theory to explore whether CEPSA's ESG transformation can achieve a sustainable development capability while protecting shareholders' interests.

### 3.2. Data Source and Analysis

The researcher incorporated DuPont analysis to conduct the current study; DuPont analysis is a financial analysis framework that breaks down the return on equity (ROE) into its parts [41,42]. The DuPont Corporation developed it in the early 20th century to assess the drivers of profitability and efficiency in a company's operations. The reason behind selecting this analysis is its ability to provide insights into the drivers of profitability and financial performance within a company.

In the current study, by breaking down ROE into parts, DuPont analysis helped the researchers to identify the specific areas contributing to a company's overall profitability and efficiency [41]. Moreover, it helped the researchers assess the impact of various factors, such as environmental performance, water management, other renewable energy, environmental management, employees, and collaboration on a company's financial performance. Thus, DuPont analysis seemed the most suited for the current study as it served as a valuable tool for understanding the drivers of the profitability and environmental performances of sample companies.

The case company was selected from Sustainalytics, where CEPSA's ESG performance was rated low risk for 3 years. It also scored well compared to MSCI, MOODY's, S&P Global, and others. These factors all indicate that CEPSA is a worthy research subject as an energy company with a significant and stable ESG transformation. The financial and non-financial data used in the case study section are secondary data from CEPSA's consolidated financial

statement, integrated management report, and Annual and Corporate Responsibility Report for 2018–2022, which are presented in Table 1 below.

**Table 1.** CEPSA's data.

| Name | Year | Report Type | Available at: |
|------|------|-------------|---------------|
| CEPSA | 2016 | CEPSA HSEQ policy | https://www.CEPSA.com/stfls/corporativo/FICHEROS/CEPSA-HSEQ-Policy.pdf (accessed on 23 October 2023) |
| CEPSA | 2018 | Annual and CSR Report | https://www.CEPSA.com/stfls/corporativo/FICHEROS/IARC2018_ENG-v2.PDF (accessed on 23 October 2023) |
| CEPSA | 2018 | Consolidated Financial Statement | https://www.CEPSA.com/stfls/corporativo/FICHEROS/CEPSA_2018_Management_Report_and_Consolidated_FFSS_English.pdf (accessed on 23 October 2023) |
| CEPSA | 2019 | Annual and CSR Report | https://www.CEPSA.com/stfls/corporativo/FICHEROS/IARC2019_ENGLISH.pdf (accessed on 23 October 2023) |
| CEPSA | 2019 | Consolidated Financial Statement | https://www.CEPSA.com/stfls/corporativo/FICHEROS/CEPSA_2019_Management_Report_and_Consolidated_FFSS_English.pdf (accessed on 23 October 2023) |
| CEPSA | 2020 | Consolidated Financial Statement | https://www.CEPSA.com/stfls/corporativo/FICHEROS/CEPSA-2020-cuentas-anuales-eng-v4.pdf (accessed on 23 October 2023) |
| CEPSA | 2020 | Integrated Management Report | https://www.CEPSA.com/stfls/corporativo/FICHEROS/CEPSA-integrated-management-report-2020-interact.pdf (accessed on 23 October 2023) |
| CEPSA | 2021 | Diversity and inclusion policy | https://www.CEPSA.com/stfls/corporativo/FICHEROS/politicas_A4_CEPSA_es_diversity_inclusion_policy_eng.pdf (accessed on 23 October 2023) |
| CEPSA | 2021 | Consolidated Financial Statement | https://www.CEPSA.com/stfls/corporativo/FICHEROS/4_NFORME_AUDITORiA_INGLeS_+_CCAA_CONSOLIDADAS_INGLeS_CEPSA_2021.pdf (accessed on 23 October 2023) |
| CEPSA | 2021 | Integrated Management Report | https://www.CEPSA.com/stfls/corporativo/FICHEROS/IGI2021ENGLISH2022.pdf (accessed on 23 October 2023) |
| CEPSA | 2022 | Integrated Management Report | https://www.CEPSA.com/stfls/corporativo/FICHEROS/igi-consolidado-2022-informe-de-verificacion-CEPSA-en.pdf (accessed on 23 October 2023) |
| CEPSA | 2022 | Consolidated Financial Statement | https://www.CEPSA.com/stfls/corporativo/FICHEROS/cuentas-anuales-consolidadas-2022-en.pdf (accessed on 23 October 2023) |

Much of the data on ESG performance and financial performance relates to the company's operations, and for reasons of data security, CEPSA does not publish first-hand trading information. It is also difficult for a single researcher to complete the collection of multi-year operational data from large multinational corporations in a short period [31,43]; therefore, secondary data use was unavoidable. The data on CEPSA consolidated financial statements and integrated management reports that have been audited by accounting firms and are highly reliable.

How to assess CEPSA's ESG performance is the key challenge in this paper. This paper will focus on analysing CEPSA's environmental performance and social responsibility performance. To solve the problem of evaluation criteria, the authors read the literature on the different dimensions of environmental performance and social responsibility performance, and then combined these with the actual business activities of CEPSA in the past five years. They summarised the factors to be considered when evaluating the environmental performance and social responsibility performance of energy-based enterprises. For example,

this paper is going to refer to Dos Santos and Pereira's (2022) [36] ESG performance scoring table, Yu et al.'s (2022) [37] evaluation methods on climate change, water resources, and human resource management in the Amazon, and Hassan et al.'s (2022) [38] findings on evaluating biodiversity. After many adjustments, the ESG performance evaluation suitable for CEPSA is summarised in Table 2.

**Table 2.** The CEPSA's ESG performance analysis structure.

| Grp. | Section | Metrics | Main Activities in CEPSA |
|---|---|---|---|
| Environmental performance | Environmental performance | Air pollution | Zero net emissions in 2050 |
| | | | Energy park |
| | Water management | Water source and consumption | Our chemical and energy parks use only fresh water that is obtained from the municipal system or other sources; Tenerife uses seawater for more than half of its water supply |
| | | Water discharge and reuse project | The La Rabida refinery (Huelva) and the Gibraltar-San Roque refinery (Cádiz) |
| | | | Position and Plan for the Use of Wastewater and Water Treatment |
| | | | A closed cycle in produced water |
| | | | Collaborating with Arcgisa to utilise sewage from cities |
| | | Water risk | At CEPSA, we use the WWF's Water Risk Filter (WRF) technology to evaluate the water risks related to our facilities |
| | Other renewable energy | Solar and wind power | The Alijar II Wind Farm in Jerez |
| | | | Europe's first network-wide installation of photovoltaic power at service stations |
| | | Biogas | The first European company producing biofuels using co-processing in 2011 |
| | | | Collaboration with a bio-methanation plant to achieve filtered soil recovery and make renewable fuels |
| | | Green hydrogen | The Andalusia Green Hydrogen Valley is Spain's largest green hydrogen park |
| | | | The continent of Europe's northern and southernmost green hydrogen corridor |
| | Environmental management | Biodiversity policy | HSEQ Policy (Health, Safety, Environment and Quality) |
| | | Ecosystem protection, habitats protected or restored | World Database on Protected Areas (WDPA) and BirdLife map |
| | | | Achievement in Madrevieja environmental station. |
| | | | Achievement in Primera de Palos Lagoon environmental station |
| | | Waste management | Reintroduction of waste into productive processes. |
| | | | Collaboration with a third party to handle the waste generated across our facilities |
| | | | Circular business (the waste hierarchy principle, the Circular Economy Board) |

**Table 2.** *Cont.*

| Grp. | Section | Metrics | Main Activities in CEPSA |
|---|---|---|---|
| Social Performance | Employee | Diversity and inclusion (gender, ethnicity, demographic, beliefs, and cultures) | Achievement of 30% female participation in leadership positions in business by 2025 |
| | | | Diversity and Inclusion Policy, standards and committees |
| | | | Concern for the employment of vulnerable groups |
| | | Career development (professional skills, salaries) | Integrated talent management model |
| | | | Integrated evaluation model |
| | | | Learning days |
| | | | Remuneration |
| | | Health and safety in the workplace | Employee health after COVID-19 |
| | | | "Zero-incident" workplace |
| | | | Healthy company program |
| | | | Health promotion programs |
| | Collaboration | Community engagement | Sumamos Energías programme |
| | | | Society relations with indigenous communities |
| | | | Social impact assessment in Colombia |
| | | Customer relationship | Developed a model to consistently measure customer experience perception |
| | | | Positive motion: new sustainable mobility models and provides the carbon footprints of products |
| | | | Commercial and clean energy production |
| | | Supplier relationship | Four-step supplier relationship management process (registration and approval, risk segment and control, performance evaluation, audit) |
| | | | wePioneer supplier recognition program |
| | | | Procurement |

## 4. Results and Discussion

### 4.1. Environmental Performance

4.1.1. Air Pollution and GHG Emission

Table 3 presents the data on CEPSA's greenhouse gas (GHG) emission changes in Scope 1 and Scope 2 from the years 2018 to 2022.

**Table 3.** CEPSA's GHG emissions changing in Scope 1 and Scope 2 from 2018 to 2022.

| GHG Emissions in Scope 1 and Scope 2 | 2022 | 2021 | 2020 | 2019 | 2018 |
|---|---|---|---|---|---|
| Exploration and Production | 168 | 178 | 285 | 468 | 461 |
| Chemicals | 857 | 1069 | 1043 | 1068 | 1102 |
| Energy Parks | 2908 | 2719 | 2802 | 3193 | 3380 |
| Commercial and Clean Energies | 1558 | 1659 | 1649 | 2047 | 1545 |
| Total (Scope 1 and Scope 2) (thousand tCO2eq) | 5491 | 5625 | 5779 | 6776 | 6488 |

Source: CEPSA's integrated management report in 2022.

Reducing greenhouse gas emissions has become particularly important in the trend of decarbonising the global economy. If people want to maintain current temperatures,

they need to ensure that GHG emissions are below the Earth's absorptive capacity [23,44]; therefore, the world needs to transition to net zero emissions.

CEPSA has set a goal of achieving net zero emissions by 2050, and to achieve this goal, CEPSA's GHG emissions in Scope 1 and Scope 2 have decreased from 6488 thousand tCO2eq in 2018 to 5491 thousand tCO2eq in 2022 (Table 3). The energy park has performed well in reducing GHG emissions, with a 14% drop over five years (Table 3). As an important part of the energy transition, CEPSA made the first attempt to convert two refineries into energy parks in 2021, and in 2022, two energy parks, Campo de Gibraltar (Cadiz) and Palos de la Frontera (Huelva), were realised for the exploitation and production of biofuels through the co-processing of vegetable oils. SAFs (Sustainable Aviation Fuels), an important product of the two energy parks, are fabricated from olivine and other plant wastes, and SAF is commonly used on commercial flights, which contributes to the decarbonisation of the European aviation industry [10,45]. At the same time, existing refineries in the energy zones are working to reduce the hydrocarbon content of crude oil to meet decarbonisation targets [46], while refining it into higher value-added products [47]. A portion of recycled and renewable feedstocks utilised in energy parks is anticipated to increase to 15% by 2030 as a crucial component of the circular economy (CEPSA integrated management report, 2022). CEPSA's energy parks in Andalusia have a leading edge in green hydrogen production. CEPSA has a straightforward route to a port and the lowest expenses for renewable energy production. Therefore, by taking advantage of the location, CEPSA can not only export, but also offer a complete decarbonisation solution to industrial, road transport, and shipping customers, as well as decarbonise hydrogen consumption at the energy park. By 2030, 70% of CEPSA's green hydrogen output will be utilised to decarbonise its clients, with the other 30% going toward meeting operational hydrogen requirements (CEPSA integrated management report, 2022).

### 4.1.2. Water Management

Figure 1 presents the data indicating the CEPSA's water withdrawal sources and highlights the changes noted from 2018 to 2020.

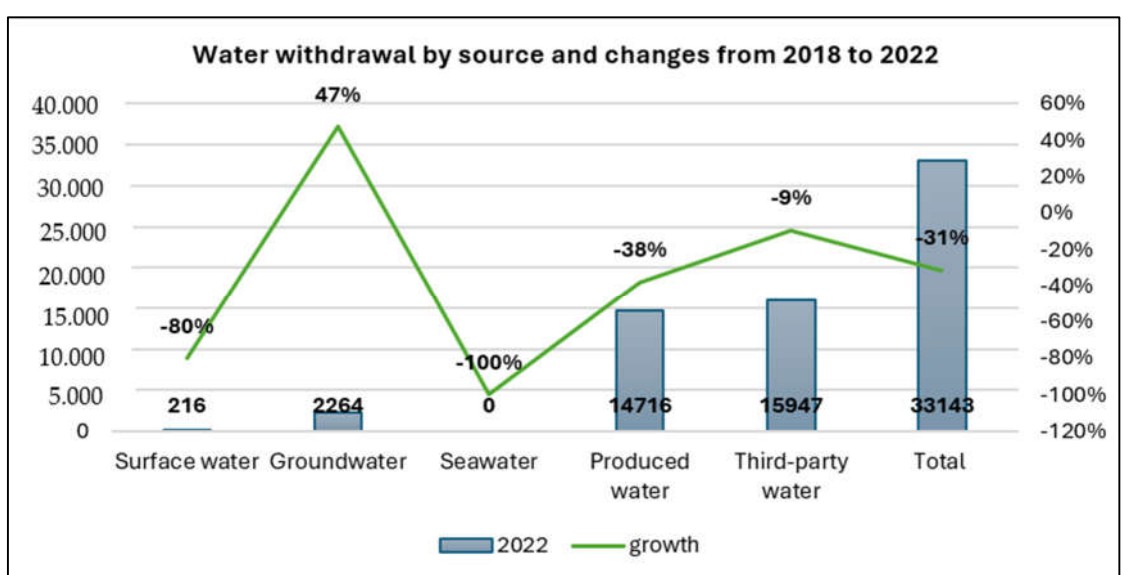

**Figure 1.** CEPSA's water withdrawal sources and changes from 2018 to 2022. Source: CEPSA's integrated management report in 2022.

Furthermore, Figure 2 presents the percentage of CEPS's freshwater recycled from 2018 to 2022.

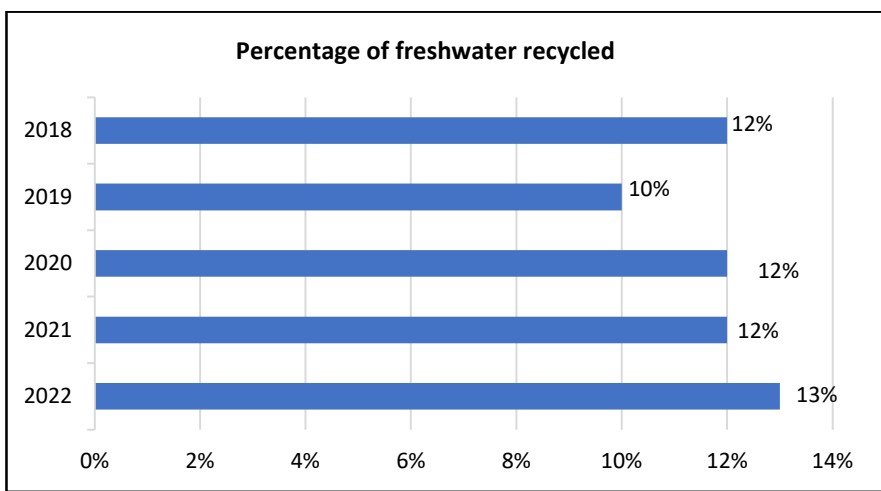

**Figure 2.** The percentage of CEPSA's freshwater recycled from 2018 to 2022. Source: CEPSA's integrated management report in 2022.

In 2018, CEPSA set out broad management objectives for the sustainable development of water resources and disclosed the sources of water withdrawal, consumption, and percentage of recycling for its main operations. To reduce water consumption in exploration and production, in 2019, CEPSA implemented multiple alternative programs. For example, increasing the number of times fresh water is utilised in the desalination process of crude oil, recycling fresh water from the pumps of fire-fighting systems, and the use of chemical solutions to bring industrial water to road spraying and cleaning standards (CEPSA's annual and corporate responsibility report, 2019). In 2022, CEPSA established a Water Committee to comprehensively manage the amount of water consumed by its operations and to set key freshwater testing indicators. The volume of freshwater used for exploration and production activities is the smallest proportion of total water withdrawal. The main sources of fresh water are also those used for domestic and industrial purposes such as groundwater and by municipal and third-party suppliers (Table 3). None of CEPSA's suppliers are involved in water-intensive products or services. CEPSA's freshwater take in water-stressed areas is expected to decrease by 25% by 2025 compared to 2019 (CEPSA's integrated management report, 2022).

To increase the recycled freshwater percentage, CEPSA improved its wastewater treatment technology in 2019, with the realisation of ultrafiltration and wastewater reuse at the treatment plants of the La Rabida refinery (Huelva) and the Gibraltar-San Roque refinery (Cadiz) (Table 4). In 2020, CEPSA proposed additional solutions for water reuse, such as using industrial wastewater for crude oil desalination, road maintenance, and dust control, and domestic wastewater for the remediation of oily waste and dilution of groundwater. By 2022, CEPSA's produced water from crude oil extraction accounted for 44.4% of the total water extracted (CEPSA's integrated management report, 2022), which is separated from the crude oil, treated to meet standards, and re-injected into the field. When additional water is needed to increase field pressure during exploration and production, the treated, produced water is returned to the field from very deep non-potable aquifers, creating a closed cycle [48]. Meanwhile, CEPSA has fully implemented urban wastewater reuse programmes at the San Roque Energy Park and the soon-to-be-built Andalusia Green Hydrogen Valley. A wastewater control plan has been established for facility management, wastewater discharges are measured, and the environmental quality of CEPSA is monitored and tested by a professional environmental company regularly.

CEPSA has also implemented a water risk assessment process for water-using facilities. Since 2017, CEPSA has used the Worldwide Fund for Nature's Water Risk Filter (WWF) to assess the environmental factors and risks associated with water withdrawals and discharges, for example, through the Water Risk Filter, in 2020, CEPSA found that the La

Rábida refinery and the exploration and production facility (BMS) in Algeria (BMS) had a serious water risk, so they developed specific improvement measures, and by 2021 the water risk in these two areas was reduced to medium risk (CEPSA's integrated management report, 2021 and 2022).

**Table 4.** The sizes of CEPSA's efforts to protect and restore habitats in 2021–2022.

| Year | Habitats Protected or Restored | Geographic Location | Size (m²)/Habitat |
|---|---|---|---|
| 2022 | Madrevieja environmental station | San Roque, Spain | 200,000 |
| | Primera de Palos lagoon | Huelva, Spain | 335,000 |
| | Las Lagunas de Muelle de las Carabelas | Huelva, Spain | 20,900 |
| | Total size (m²) | | 555,900 |
| 2021 | Madrevieja environmental station | San Roque, Spain | 200,000 |
| | Primera de Palos lagoon | Huelva, Spain | 335,000 |
| | La Rábida | Huelva, Spain | 19,700 |
| | Total size (m²) | | 554,700 |

Source: CEPSA's integrated management report, 2021.

### 4.1.3. Other Renewable Energy Management

CEPSA officially entered the renewable energy business in 2018 with the acquisition of the Alijar II Wind Farm in Jerez. This wind farm has helped them reduce greenhouse gas emissions by 320 million tonnes per year. In 2019, CEPSA installed solar panels at 10 service stations, and by 2021, CEPSA partnered with Redexis to build Europe's first photovoltaic network at service stations (CEPSA's integrated management report, 2021). CEPSA's solar panels have been installed at 80 service stations, where they power the stations during the day, while the remaining power is fed into the grid to optimise the overall energy efficiency of the facility, making the stations 100% renewable.

For biofuels, CEPSA endeavours to reduce its reliance on external sources. In 2011, CEPSA became the first company in Europe to produce biofuels through co-processing. Today, they produce more than 1 million tonnes of biofuels (CEPSA's integrated management report, 2021), providing a basis for reducing carbon emissions. Since 2019, CEPSA has tried to integrate biofuel production into the production chain, and now they are breaking through the collaborative processing of second-generation material. At the same time, the Roque bioenergy plant has achieved 100 percent filtered soil recycling in cooperation with a bio methanation plant of HTN (CEPSA's integrated management report, 2021). Between 2023 and 2030, CEPSA expects to ally with biogas producers to make renewable fuels from plants, forests, and animal manure. The use of biogas will promote the decarbonisation of all CEPSA's industrial facilities in the Iberian Peninsula.

CEPSA's main activities are all related to crude oil, so they are constantly working to optimise their production processes to make the best use of all the sub-products generated by processing crude oil. Meanwhile, CEPSA seeks to realise the energy transition through green hydrogen, which is produced by processing fully sustainable and renewable resources that do not produce carbon emissions. It can be utilised in a variety of industries, including transportation, aviation, and electrification. To this purpose, CEPSA 2022 has announced the development of the Andalusia Green Hydrogen Valley, which is anticipated to create 300,000 tonnes of green hydrogen annually and will be the largest green hydrogen park in Spain (CEPSA's integrated management report, 2021). By favour of the EU's RePower policy, CEPSA will also construct a primary green hydrogen corridor between northern and southern Europe.

Considering the high carbon intensity associated with crude oil processing, CEPSA has also implemented some compensatory measures. For example, CEPSA accrues more carbon allowances to compensate for the $CO_2$ emissions generated by its production activities.

These carbon allowances come from participation in a rainforest restoration project in Brazil, which aims to avoid the deforestation of more than 4000 hectares of rainforest and compensate for 1.4 million tonnes of $CO_2$ emissions (CEPSA's annual and corporate responsibility report, 2019).

### 4.1.4. Biodiversity Policy and Ecosystem Protection

Table 4 presents the sizes of CEPSA's efforts to protect and restore habitats in the years 2021 and 2022.

CEPSA has made a lot of efforts to protect biodiversity and ecosystems and has achieved some results. Early in 2009, CEPSA set up internal standards for the protection of biodiversity; in 2014, it initiated an assessment and protection process and set up a fund to provide financial support (EPSA's annual and corporate responsibility report, 2018). Later, CEPSA issued a policy aimed at environmental protection, the 'HSEQ Policy' (Health, Safety, Environment and Quality), which regularly identifies and evaluates the ecological impact of its operations (CEPSA HSEQ policy, 2016). CEPSA refrained from conducting research or production tasks in regions designated as UNESCO World Heritage Sites. To assess the environmental impact of each of its operations more comprehensively, CEPSA has developed in-house analysis tools using two major databases: the World Database on Protected Areas (WDPA), which helps CEPSA to effectively identify protected areas that overlap, are adjacent to, or are near the company's operations, and BirdLife map, which helps CEPSA identify bird species and areas of intense bird activity within its operating area. After years of work, in 2022, CEPSA discovered new species—egrets, crabeater herons, cattle egrets, and spoonbills—at the environmental station in Primera de Palos lagoon (Palos de la Frontera, Spain). To protect species diversity and reduce the risk of fire, CEPSA has also established protected areas and has been working to restore and maintain the ecosystems in them.

### 4.1.5. Waste Management

Figure 3 presents CEPSA's waste management plan for both hazardous and non-hazardous waste for 2018–2022, as outlined in Table 5.

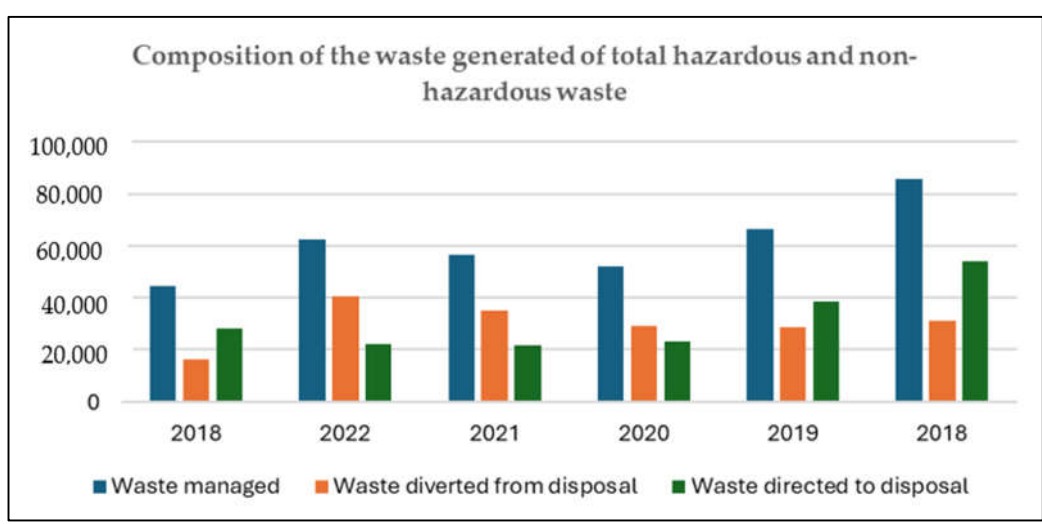

**Figure 3.** CEPSA's waste management of total hazardous and non-hazardous waste from 2018 to 2022. Source: CEPSA's integrated management report, 2022.

CEPSA has emphasised the importance of waste recycling and waste reuse in the production process since 2018. The total amount of waste managed by CEPSA in 2019 was 66,616 tonnes, of which 28,366 tonnes were recycled and reused (Table 3). CEPSA has also set criteria for the classification of waste: hazardous, non-hazardous, or municipal waste. Working with third parties is a vital aspect of CEPSA's waste management strategy because

authorised processors can perform the proper exterior treatment for multiple waste kinds, boost recycling rates, and reduce landfill usage. To make a holistic change in CEPSA's resource-consuming mode of operation, they proposed to shift from a linear economy to a circular economy model back in 2018 (EPSA's annual and corporate responsibility report, 2018). At that time, it was limited only to waste management, but by 2022, CEPSA has been working on multiple angles of integrated degree management by reducing the use of raw materials, increasing the use of renewable materials, and reducing the generation of waste and recycling waste (CEPSA's integrated management report, 2022). A Circular Economy Committee has also been established to find recyclable alternatives to waste and raw materials in production.

**Table 5.** Two-factor analysis of CEPSA's ROE from 2018 to 2022.

| **2018** | |
| --- | --- |
| ROE (1) = ROA (2018) $*$ EM (2018) | 15.21% |
| ROE (2) = ROA (2018 + 1) $*$ EM (2018) = ROA (2019) $*$ EM (2018) | 14.17% |
| ROE (3) = ROA (2019) $*$ EM (2018 + 1) = ROA (2019) $*$ EM (2019) | 15.73% |
| The impact of the increase in EM on the ROE in 2019 was as follows: | |
| ROE (3) − ROE (2) | 1.56% |
| The impact of the increase in EM on the ROE in 2019 was as follows: | |
| ROE (3) − ROE (2) | 1.56% |
| **2019** | |
| ROE (1) = ROA (2019) $*$ EM (2019) | 15.73% |
| ROE (2) = ROA (2019 + 1) $*$ EM (2019) = ROA (2020) $*$ EM (2019) | −19.38% |
| ROE (3) = ROA (2020) $*$ EM (2019 + 1) = ROA (2020) $*$ EM (2020) | −22.71% |
| The impact of the decrease in ROA on the ROE in 2020 was as follows: | |
| ROE (2) − ROE (1) | −35.10% |
| The impact of the increase in EM on the ROE in 2020 was as follows: | |
| ROE (3) − ROE (2) | −3.33% |
| **2020** | |
| ROE (1) = ROA (2020) $*$ EM (2020) | −22.71% |
| ROE (2) = ROA (2020 + 1) $*$ EM (2020) = ROA (2021) $*$ EM (2020) | 15.24% |
| ROE (3) = ROA (2021) $*$ EM (2020 + 1) = ROA (2021) $*$ EM (2021) | 16.78% |
| The impact of the increase in ROA on the ROE in 2021 was as follows: | |
| ROE (2) − ROE (1) | 37.94% |
| The impact of the increase in EM on the ROE in 2021 was as follows: | |
| ROE (3) − ROE (2) | 1.54% |
| **2021** | |
| ROE (1) = ROA (2021) $*$ EM (2021) | 16.78% |
| ROE (2) = ROA (2021 + 1) $*$ EM (2021) = ROA (2022) $*$ EM (2021) | 22.81% |
| ROE (3) = ROA (2022) $*$ EM (2021 + 1) = ROA (2022) $*$ EM (2022) | 23.42% |
| The impact of the increase in ROA on the ROE in 2022 was as follows: | |
| ROE (2) − ROE (1) | 6.04% |
| The impact of the increase in EM on the ROE in 2022 was as follows: | |
| ROE (3) − ROE (2) | 0.61% |

Source: CEPSA's Consolidated financial statements.

*4.2. Social Performance*

4.2.1. Employee Management in Diversity and Inclusion

CEPSA's professional management team is an important resource, uniting employees with shared values and motivating them in their work to help them achieve their career goals. Employee diversity and inclusion is also a key focus for CEPSA, and they are committed to having 30% of leadership positions within the company filled by women by 2025 (CEPSA's Integrated Management Report, 2020).

To build a diverse and inclusive work environment, CEPSA signed the Diversity Charter in 2019, and the new 'diversity and inclusion policy' published in 2021 also supports CEPSA's dedication to protecting all workers' human rights. CEPSA respects the human rights of all employees and prevents discrimination based on gender, ethnic background, faith, religion, age, and any other circumstance (The Board of Directors of CEPSA, 2021). To this end, they have also set up a diversity and inclusion committee to address challenges and opportunities, select best practice processes, track project progress, and achieve a diverse and inclusive culture. CEPSA has always been concerned about the employment situation of disadvantaged groups, and they proposed to create an environment that favours the inclusion of people with disabilities in 2018. In 2022, CEPSA created another organisation internally, called Capaz, which is dedicated to speaking out on behalf of people with disabilities. Other organisations like this are Anexa (an organisation that promotes the equality of awareness between men and women) and Equal (an organisation that raises the awareness of gender and sexual orientation and promotes LGBTI inclusiveness) (CEPSA's integrated management report, 2022). Additionally, EPSA collaborates with workers' legal counsel to establish collaborations that are devoid of employment discrimination and dedicated to fostering workplaces that uphold human rights.

4.2.2. Employee Management in Career Development

CEPSA strives to attract and retain talent to upgrade its skills, and to achieve this, they introduced the integrated talent management model in 2018, the key to which is the 'Talent Call' programme. The programme attracts high-quality talent by recruiting students and graduates and providing them with a first work experience. In the same year, CEPSA signed an industry sustainability agreement, which included a dual vocational training programme to ensure the supply of talent (EPSA's annual and corporate responsibility report, 2018). In 2020, CEPSA further refined the model, proposing an integrated talent management approach in four dimensions: Talent Cal, talent recruitment for attracting the best recent graduates; Internal Mobility, facilitating talent mobility for skill development; Talent in Motion, ensuring that employees acquire sufficient professional skills; and Unleash Your Energy, enabling employees to find the right leadership style in the company and unleash their energy (CEPSA's integrated management report, 2022).

In 2022, CEPSA proposed for the first time the creation of an integrated evaluation model for employees, which will help organisations to continuously monitor the employees' skill development (CEPSA's integrated management report, 2022). The model consists of the following areas. A MIDE system to assesses the discrepancy between personal performance and objectives, which is not limited to performance indicator assessments but will also assess the leader's leadership style and the leader's performance. Talent committees assess department heads and senior skilled personnel, evaluating potential opportunities and looking for ways to address future challenges. Lastly, succession plans assess key positions and their successors and develop career plans (which are presented in Figure 4) for them to ensure that they will be able to successfully take over key positions.

It is evident from Figure 4 that CEPSA has set up a systematic training program for its employees. In 2022, they officially designated the third Friday of every month as a learning day, on which all employees are required to receive three hours of training. Employees of CEPSA are also provided a monthly schedule of e-learning programs covering a range of subjects. Among these are ESG, new renewable energy sources, digital skills, diversity, and inclusion. These training courses serve the employees' work and consist of 70%

practice and experience, 20% coaching, and 10% training. They also provide a LinkedIn Learning training catalogue for all staff so that staff can acquire new skills through self-study (CEPSA's integrated management report, 2022). In terms of compensation reform, CEPSA is proposing to link employee compensation to ESG performance from 2022. In the same year, CEPSA's ESG criteria accounted for 20–25% of the company's annual targets.

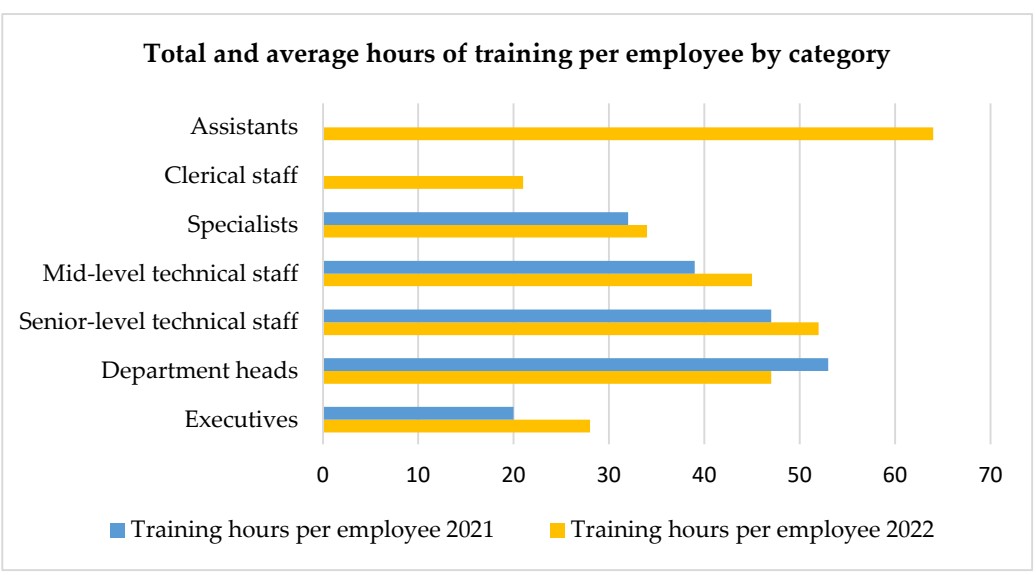

**Figure 4.** CEPSA's employee training hours in different categories. Source: CEPSA's integrated management report, 2022.

### 4.2.3. Employee Management in Workplace Health and Safety

The security management of CEPSA places a high priority on employee safety and well-being. Immediately after the COVID-19 outbreak in 2020, CEPSA initiated a contingency plan, organising representatives from different areas of the business to form a committee to work together to deal with the outbreak. They are responsible for both monitoring the health of employees, their families, and other stakeholders, as well as analysing and dealing with outbreaks caused by COVID-19 and preparing for recovery from the pandemic (CEPSA's integrated management report, 2020). In 2021, CEPSA introduced reforms to address COVID-19 and employee health and safety, including more involvement of the Crisis Management Committee in the management of the enterprise, the reduction of the risk of COVID-19 transmission through telecommuting, medical support for infected workers, the detection of the evolution of COVID-19, the aggregation of the information it collects for professionals, etc. (CEPSA's integrated management report, 2021).

The number of CEPSA employee injuries still had one person in 2020 and 2021, but it became zero in 2022. This change indicates that a healthy and safe workplace is also important for the ESG performance of an organisation, and in 2018, CEPSA set a target of 'zero accidents'. To achieve this goal, they have improved the safety of their facilities and have improved the maintenance and control of their equipment. They have also stepped up their efforts to promote a safety culture, and in 2018, CEPSA published an Ethics and Compliance Channel Policy on its website, which establishes a complaint procedure for unhealthy and unsafe workplaces. By 2022, CEPSA had also set up specific procedures to help employees reduce the risks inherent in performing specific tasks, based on the requirements of the Spanish Institute for Occupational Health and Safety (INSST).

In 2017, CEPSA launched the "Healthy Company program", which focuses on the health and well-being of employees outside of work. For example, in 2022, the employee emotional support program worked with specialist external services to help employees achieve work–life balance or run employee emotional support programs to reduce the risk of mental illness (CEPSA's integrated management report, 2022).

4.2.4. Collaboration with Community

In the process of oil and gas exploration and production, CEPSA will face many community issues. CEPSA, as early as 2018, has carried out co-operation projects with the communities in the main operating areas. Examples include providing Spanish teaching assistants to the second-largest school in the Algerian region or helping Canadian disadvantaged groups improve their quality of life (EPSA's annual and corporate responsibility report, 2018). However, these projects are mostly point-to-point assistance, with fragmentation between projects and a lack of systematicity. To maximise the integration of its renewable energy installations into their surroundings and generate prospects for the economic growth of local communities, CEPSA established the Sumamos Energias initiative in 2022. (CEPSA's integrated management report, 2022). During exploration and production operations, CEPSA sends experts to help community members increase project participation and focus on specific stakeholders.

CEPSA endeavours to build transparent relationships with the communities in which it operates. The number of senior managers from local community increased from 92 person to 140 person, and in 2022, CEPSA released its Social Relations Manual, which provides guidelines for engaging with partners and indigenous communities. When their operations involve these communities, transparency of information and equal communication are guaranteed to drive smooth operations. Regular meetings and round tables are held to involve indigenous communities in projects and provide them with opportunities for investment and employment. Finally, the company also regularly evaluates its impact on indigenous communities across three key indicators—social inclusion, poverty, and capacity—and in 2022, after comparing CEPSA's direct impact on the local community in the Caracara block (Colombia) with the indirect impact on the indigenous community, they found that CEPSA made a positive contribution on all three key indicators (CEPSA's integrated management report, 2022).

4.2.5. Collaboration with Customer

In 2019, CEPSA developed a model that allows for the continuous measurement of customer experience. It generates net promoter scores by establishing satisfaction metrics through customer satisfaction surveys, collects and analyses changes in the metrics to understand customer needs and expectations, and then improves services. In this plan, CEPSA is passive only in providing services; they change their services and products according to the change in customer needs (EPSA's annual and corporate responsibility report, 2019). However, as the customer satisfaction data improve, CEPSA proposed a 'positive motion' programme in 2022, which aims to provide customers with the means to respond effectively to decarbonisation (CEPSA's integrated management report, 2022). To promote customer mobility and road transport decarbonisation, CEPSA is developing one of the largest sustainable mobility ecosystems in Spain and Portugal. Customers will be able to charge their electric vehicles in less than 20 min. by supplying a mobile charging network for business-to-business clients to ease their transition to environmentally friendly mobility. CEPSA encourages the use of hydrogen for heavy-duty road transportation, converting 1800 currently existing service stations in Portugal and Spain into digital, incredibly convenient sites that provide clients with top-notch dining and resting options. CEPSA will also provide customers with information on the carbon footprint of their products at different stages of the life cycle, thus helping them to manage their carbon footprint and decarbonise sooner. CEPSA estimates reducing the carbon intensity of the electricity it sells to final users by 15% to 20% by 2030.

CEPSA's new strategy puts customers at the centre of decarbonisation by offering a wide range of clean products, including wholesale B2B, aviation, lubricants, bitumen, LPG, natural gas, and electricity, and by creating a value chain around biofuels, hydrogen, and renewable energy (solar and wind) businesses to help customers build decarbonisation solutions (CEPSA's integrated management report, 2022). Between 2021 and 2022, CEPSA entered strategic partnerships with Iberia and Iberia Express and Seville Airport to develop

and mass produce Sustainable Aviation Biofuels (SAFs) to advance the decarbonisation of the aviation industry, using waste, used oils, and second-generation biofuels.

### 4.2.6. Collaboration with the Supplier

Before 2020, CEPSA paid little attention to supplier relationships; however, since 2020, managers have realised the importance of supplier management for business sustainability. As a result, CEPSA has developed a four-step supplier relationship management process over the past three years (CEPSA's integrated management report, 2020). Registration and Approval: To guarantee that suppliers accept the demands of the business, the first step is to determine the obligations that they must fulfil. CEPSA also measures the risk level of suppliers, so all registered suppliers must complete an ESG questionnaire. The company will decide whether to work with suppliers based on their ESG scores. Additionally, the evaluation results are posted on the procurement portal, where they are automatically added to the tender list for the consideration of bids each year. The second step is Risk Segmentation and Control. CEPSA categorises suppliers based on the risk level and type calculated from the ESG scores and identifies key suppliers. This step also adds the requirement for managers to assess human right risks and due diligence. The third step is Performance Evaluation. It is an assessment of suppliers that have worked together several times and covers product quality, execution, and ESG performance. Every year, 99% of suppliers go through at least one performance evaluation. The final step is the AUDIT, which is a regular supplier audit that ensures that suppliers meet CEPSA's internal corporate requirements and internationally recognised ESG standards. During audits, CEPSA works with suppliers to rectify non-compliant items and encourages suppliers to close non-compliant items.

CEPSA also offers training programmes for suppliers and purchasers. Since 2019, CEPSA has offered ESG training for suppliers who share the same values through the WePioneer supplier recognition programme. Meanwhile, since 2021, companies have added ESG training for purchasers, and after learning from the lessons of COVID-19, which led to unstable supplies and price fluctuations from international suppliers, CEPSA encourages procurers to purchase from local suppliers.

### 4.3. Financial Performance

4.3.1. Two-Factor Analysis of Return on Equity (ROE)

Figure 5 presents a line chart of CEPSA's indicator change from 2018 to 2022.

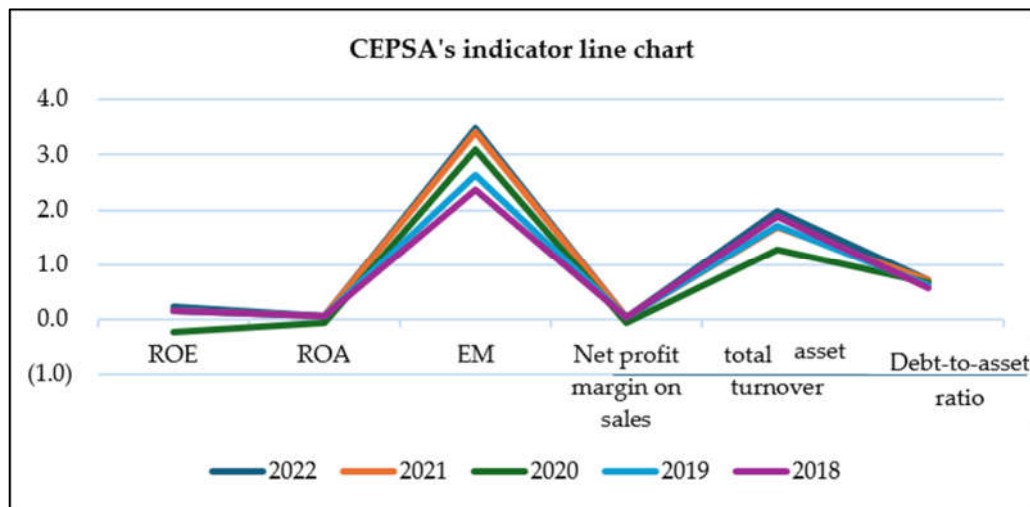

**Figure 5.** A line chart of CEPSA's indicator change from 2018 to 2022. Source: CEPSA's Consolidated financial statements.

Additionally, Table 5 provides comprehensive two-factor analysis results that reveal valuable insight into CEPSA's financial performance, highlighting the company's commitment to transparency and sound financial management of the company's return on equity (ROE) during 2018–2022.

An analysis of the above calculations found an overall increase in the return on assets (ROA), and equity multiplier (EM) from 2018 to 2022, so the return on equity (ROE) shows an increasing trend in the same period. The change in ROA is indicative of CEPSA's overall smooth running in recent years, with a significant decline in 2020 but a rapid recovery in 2021 and 2022 (Figure 5). This indicates that CEPSA's asset utilisation and profitability are strong and are expected to maintain their growth trend in the future [30]. EM is a measure of the proportion of firms that are financed with equity. CEPSA's EM has been rising steadily and at a faster rate over the last five years, implying that the reliance on debt financing within firms is gradually increasing. This finding has also been supported by the change in the debt-to-asset ratio, which has increased year on year over the last five years, increasing the debt risk faced by companies and investors [18]. It is worth noticing that the ROA declined in 2019 and 2020.

According to the report, revenue decreased by 3.58% in 2019 and decreased by 33.9% in 2020, while total assets still kept an increasing trend in 2019 and decreased by 10.93% only in 2020 (CEPSA's Consolidated financial statements, 2020 and 2019). The current asset decreases of 16.35% is the main factor resulting in the total asset decrease, but its decrease level is still lower than the revenue decrease in 2020. These changes are further evidence that it is CEPSA's profitability declining and not simply a decline in earnings due to a reduction in current assets. The main reason for the increase in EM is the decline in shareholder equity from EUR 5.3 billion to EUR 4.028 billion and the more than tripling of current liability between 2018 and 2022 (Figure 5), although there also was a sudden decline in current liability in 2020. It fell from EUR 3.576 billion to EUR 2.656 billion (CEPSA's Consolidated financial statements, 2022). As an important indicator of the level of corporate debt, the trend in debt–asset ratios is consistent with the EM. With the debt level rising, CEPSA needs to evaluate its solvency. Compared with non-current liability, CEPSA's current liability growth rate is rapid, which means the company needs to pay more interest, and the liquidity pressure rises annually.

4.3.2. Three-Factor Analysis of Return on Equity (ROE)

Table 6 presents the results of a three-factor analysis of CEPSA's return on equity (ROE) between 2018 and 2022.

**Table 6.** Three-factor analysis of CEPSA's return on equity (ROE) from 2018 to 2022.

| 2018 | |
| --- | --- |
| ROE (1) = NPMS (2018) * TAT(2018) * EM(2018) | 15.21% |
| ROE (2) = NPMS (2018 + 1) * TAT (2018) * EM (2018) = NPMS (2019) * TAT (2018) * EM (2018) | 15.58% |
| ROE (3) = NPMS (2019) * TAT (2018 + 1) * EM (2018) = NPMS (2019) * TAT (2019) * EM (2018) | 14.17% |
| ROE (4) = NPMS (2019) * TAT (2019) * EM (2018 + 1) = NPMS (2019) * TAT (2019) * EM (2019) | 15.73% |
| The impact of the increase in NPMS on the ROE in 2019 was as follows: | |
| ROE (2)-ROE (1) | 0.37% |
| The impact of the decrease in TAT on the ROE in 2019 was as follows: | |
| ROE (3)-ROE (2) | −1.41% |
| The impact of the increase in EM on the ROE in 2019 was as follows: | |
| ROE (4)-ROE (3) | 1.56% |

**Table 6.** *Cont.*

| 2019 | |
| --- | --- |
| ROE (1) = NPMS (2019) * TAT (2019) * EM (2019) | 15.73% |
| ROE (2) = NPMS (2019 + 1) * TAT (2019) * EM (2019) = NPMS (2020) * TAT (2019) * EM (2019) | −26.12% |
| ROE (3) = NPMS (2020) * TAT (2019 + 1) * EM (2019) = NPMS (2020) * TAT (2020) * EM (2019) | −19.38% |
| ROE (4) = NPMS (2020) * TAT (2020) * EM (2019 + 1) = NPMS (2020) * TAT (2020) * EM (2020) | −22.71% |
| The impact of the decrease in NPMS on the ROE in 2020 was as follows: | |
| ROE (2)-ROE (1) | −41.85% |
| The impact of the decrease in TAT on the ROE in 2020 was as follows: | |
| ROE (3)-ROE (2) | 6.75% |
| The impact of the increase in EM on the ROE in 2020 was as follows: | |
| ROE (4)-ROE (3) | −3.33% |
| **2020** | |
| ROE (1) = NPMS (2020) * TAT (2020) * EM (2020) | −22.71% |
| ROE (2) = NPMS (2020 + 1) * TAT (2020) * EM (2020) = NPMS (2021) * TAT (2020) * EM (2020) | 11.52% |
| ROE (3) = NPMS (2021) * TAT (2020 + 1) * EM (2020) = NPMS (2021) * TAT (2021) * EM (2020) | 15.24% |
| ROE (4) = NPMS (2021) * TAT (2021) * EM (2020 + 1)= NPMS (2021) * TAT (2021) * EM (2021) | 16.78% |
| The impact of the increase in NPMS on the return on ROE in 2021 as follows: | |
| ROE (2)-ROE (1) | 34.23% |
| The impact of the increase in TAT on the ROE in 2021 was as follows: | |
| ROE (3)-ROE (2) | 3.71% |
| The impact of the increase in EM on the ROE in 2021 was as follows: | |
| ROE (4)-ROE (3) | 1.54% |
| **2021** | |
| ROE (1) = NPMS (2021) * TAT (2021) * EM (2021) | 16.78% |
| ROE (2) = NPMS (2021 + 1) * TAT (2021) * EM (2021) = NPMS (2022) * TAT (2021) * EM (2021) | 19.27% |
| ROE (3) = NPMS (2022) * TAT (2021 + 1) * EM (2021) = NPMS (2022) * TAT (2022) * EM (2021) | 22.81% |
| ROE (4) = NPMS (2022) * TAT (2022) * EM (2021 + 1) = NPMS (2022) * TAT (2022) * EM (2022) | 23.42% |
| The impact of the increase in NPMS on the ROE in 2022 was as follows: | |
| ROE (2)-ROE (1) | 2.49% |
| The impact of the increase in TAT on the ROE in 2022 was as follows: | |
| ROE (3)-ROE (2) | 3.54% |
| The impact of the increase in EM on the ROE in 2022 was as follows: | |
| ROE (4)-ROE (3) | 0.61% |

Source: CEPSA's Consolidated financial statements.

EM was analysed in the last part; therefore, this part will focus on the total asset turnover (TAT) and net profit margin on sales (NPMS). Small changes in TAT and NPMS occurred between 2018 and 2022. The TAT shows an increase from 1.87 to 1.98 (Figure 5), while the NPMS shows a slight decline from 3.41% to 3.38% in these five years (Figure 5). A sharp decline in the NPMS also occurred in 2020. TAT is a measure of a company's ability to generate revenue from existing assets, a rising TAT represents a greater sales ability. The TAT rose very little between 2018 and 2022, but this does not mean that CEPSA's sales capacity stayed stagnant. Over the five years, CEPSA's net sales growth rate was 35.35%, but total assets also grew at a rate of 28.17% (CEPSA's Consolidated financial statements,

2022 and 2018), meaning that the business spent most of its income on investment and reproduction.

Observing changes in the NPMS is useful for investors to assess how well managers are controlling their operating costs. A decline in the NPMS indicates a weakness in operating cost control. The reason for the slightly lower performance of the NPMS in 2022 compared to 2018 is that the NPMS showed a sharp decline in 2020 and remained in a gradual recovery until 2022 (Table 6). At the same time, the NPMS was on an upward trend in 2021 and 2022, so it is not possible to rely solely on the comparison between 2022 and 2018 to conclude that CEPSA's ability to control operating costs has declined.

### 4.3.3. Comparison of ROA, ROE, and EM between CEPSA and Shell

To test the impact of CEPSA's ESG performance improvement based on its financial performance, the authors compared the changes between Shell [49,50] and CEPSA in terms of key financials such as ROE, ROA, and EM over the period 2018–2022 which is presented in Table 7.

**Table 7.** Comparative data of ROA, ROE, and EM of CEPSA and Shell (2018–2022).

| Item/Year | CEPSA | | | | | Shell | | | | |
|---|---|---|---|---|---|---|---|---|---|---|
| | **2022** | **2021** | **2020** | **2019** | **2018** | **2022** | **2021** | **2020** | **2019** | **2018** |
| ROE | 23.42% | 16.78% | −22.71% | 15.73% | 15.21% | 22.26% | 11.77% | −13.58% | 8.63% | 11.80% |
| ROA | 6.71% | 4.93% | −7.35% | 5.97% | 6.40% | 9.68% | 5.10% | −5.68% | 4.06% | 5.99% |
| EM | 3.49 | 3.40 | 3.09 | 2.64 | 2.38 | 2.30 | 2.31 | 2.39 | 2.12 | 1.97 |
| net profit margin on sales | 0.03 | 0.03 | −0.06 | 0.03 | 0.03 | 0.11 | 0.08 | −0.12 | 0.05 | 0.06 |
| total asset turnover | 1.98 | 1.67 | 1.27 | 1.71 | 1.88 | 0.87 | 0.67 | 0.48 | 0.87 | 0.99 |
| debt to asset ratio | 0.71 | 0.71 | 0.68 | 0.62 | 0.58 | 0.57 | 0.57 | 0.58 | 0.53 | 0.49 |

Source: CEPSA's Consolidated financial statements and Shell's annual report and accounts.

The table above demonstrates that the ROEs of CEPSA and Shell had an overall upward trend over the five-year period, but Shell's ROE declined from 11.8% to 8.36% between 2018 and 2019 and rose to 22.26% in 2022. Both of their ROEs fell extremely fast to negative values in 2020. Changes in net income were the main cause of volatility in Shell's ROE, with net income declining by 31.26% during 2018–2019 and rising by 107.82% in 2021–2022. The decline in revenue during 2018–2019 was primarily caused by integrated gas and downstream consumers. Shell natural gas revenues are highly volatile due to regional natural gas prices. Compared to 2018, natural gas prices per million British thermal units (MMBtu) fell by 18% in the United States, 43% in Europe, and 43% in Asia–Pacific in 2019, which significantly impacted Shell's revenue from integrated gas sales. Meanwhile, operating expenses for integrated gas rose by 10.85% despite declining operating expenses in all other businesses, with more than 50% of the increase occurring due to the added renewable energy business (Shell's 2019 annual report) [50]. Commodity trading is an important part of Shell's upstream, integrated natural gas, and downstream businesses, and changes in commodity prices are one of the main risks for variability in revenues. The decline in downstream revenues in 2019 was primarily due to declines in realised chemicals and refining and trading margins and a decline in fair value-related gains on commodities. The fast rise in the revenue of Renewables and Energy Solutions between 2020 and 2022 is the main reason for the rise in net income. In 2021, Renewables and Energy Solutions' revenue rose by 93.79%, and in 2022 by 121.41%. Shell's revenues from its upstream and integrated gas businesses are currently growing despite the ongoing downturn in oil and gas prices. For the foreseeable future, the profitability of this segment of the business could decline or lose money. In response to this risk, Shell has developed the Renewables and Energy Solutions business in 2020. The business is primarily involved in renewable energy generation, natural gas, the trading of carbon credits, and the production and marketing

of hydrogen [45–47]. The business also includes the development of carbon capture and storage centres, while Shell Venture was established to accelerate the pace of Shell's ESG transformation by investing in companies that are transforming energy and transport (Shell's 2022 annual report) [50]. Shell invested $38.07 million in Carbonext in 2022 to help them achieve their goal of offsetting 120 million tonnes of $CO_2$ per year by 2030 (KnowESG, Shell plc, 2022). Both CEPSA and Shell's ROAs showed an upward trend over the five years, but they both experienced slight declines in 2018–2019. CEPSA's ROA declined by 0.43% and Shell's ROA declined by 1.93%. Shell's ROA was slightly lower than CEPSA in 2019, but in 2022, Shell's ROA reached 9.68%, exceeding CEPSA's 6.71%. This indicates that Shell has a stronger profitability in using its assets compared to CEPSA in 2022. The equity multiplier for both companies has risen consistently over the five years. CEPSA has risen by 47.01%, and Shell has risen by 16.71%. This indicates that both companies have increased the proportion of equity financing. Concerning the debt-to-asset ratio, CEPSA's debt-to-asset ratio has been above 0.5 for all five years and has risen by 23.25 per cent, which shows that debt financing is still the main source for CEPS Inc. However, Shell's debt-to-asset ratio has been maintained at around 0.5 and is even showing a small decline in 2021–2022, which indicates that Shell's debt and equity financing is balanced [47].

### 4.3.4. Change in Revenue

CEPSA's operating income declined sharply between 2019 and 2020 but recovered quickly and grew steadily in 2021. The decline in revenues is partly due to lower crude oil prices and slower domestic demand. Although the start-up of the Sarb and Umm Lulu fields in Abu Dhabi increased volumes in the exploration and production areas, this was partially offset by lower average selling prices. This was also partly due to the continued plunge in oil prices because of the COVID-19 outbreak, which led to further declines in domestic and international demand. Volumes in the exploration and production sector, refineries, electricity and gas, and aviation fuels were all significantly affected, and profits fell sharply, but CEPSA's petrochemical sector continued to perform well as CEPSA petrochemical plants produce chemicals that are useful in fighting disease and stopping the spread of disease (CEPSA's Consolidated financial statements, 2022), such as LAB and phenol.

In 2021, crude oil prices rebounded as market demand for petroleum products gradually recovered, with the impact of COVID-19 diminishing, but still did not return to pre-pandemic levels. Aviation fuel volumes remained low due to global travel constraints. In 2022, in line with the sustainability transition, CEPSA launched the positive motion program, which on the one hand, increased the average utilisation of installed refinery capacity, and on the other hand, offered fuel discounts to customers, all of which were effective in boosting volume growth. Volume growth was also driven by increased market demand, with CEPSA's power generation up 7% in 2022 compared to 2021 (CEPSA's Consolidated financial statements, 2022). The gas and power business also saw an increase in total revenues due to the increase in natural gas prices.

### 4.3.5. Change in Liability

CEPSA's total liabilities have risen by 57.98% over the five years. Before 2020, the acquisition of petrol stations and the purchase of new equipment for the exploration and production areas led to a significant increase in other non-trade payables, while the lack of an integrated sustainable management plan and high greenhouse gas emissions from production processes resulted in $CO_2$ emissions exceeding the nationally mandated $CO_2$ emission allowance (CEPSA's Consolidated financial statements, 2018 and 2019). Therefore CEPSA had a large provision for environmental activities in 2018 and 2019. In both the consolidated income statement and the income statement's short-term reserve, these charges are shown as other operating expenses. After 2020, in reaction to the liquidity shortfall caused by COVID-19, CEPSA signed a new three-year loan facility. A voluntary moratorium on employment for retirees was launched, and liabilities rose because of the

choice to dismantle the refinery in Tenerife without any social or environmental incidents, due to the strict observance of the abandonment plan and the principles of the circular economy, as well as the provision for the possible costs of subsequent soil remediation.

4.3.6. Change in Investment

CEPSA has been actively collaborating with other companies over the past five years to become a leader in the transport and sustainable energy sectors. In the chemical business, CEPSA invested more than EUR 46 million in 2022 and 2021 for China phenol production plant renovation, plant environmental upgrading, and decarbonisation (CEPSA's Consolidated financial statements, 2022 and 2021). CEPSA is also committed to expanding other business's renewable energy portfolios in Spain and Portugal in partnership with Abu Dhabi's future energy company, Masdar. CEPSA and Redexis have invested in constructing the largest network of Vehicle Natural Gas (VNG) refuelling stations, providing additional solutions for sustainable transport energy.

In 2022, CEPSA launched the 'positive motion' program. Before the program was launched, CEPSA renovated its chemical plant in Puente Mayorga, consolidating its leading position in the LAB manufacturing sector. Using new technologies has improved plant efficiency and sustainability and reduced operating costs. In the refining sector, the continuous improvement of refinery equipment improves efficiency and conversion rates, minimises environmental impact and ensures the safety of employees. CEPSA also added Europe's largest hydrocracking unit to its refinery in La Rábida (Huelva). With 'positive motion' in place, CEPSA established the first green hydrogen corridor between northern and southern Europe, advanced its network of ultra-fast electric vehicle chargers (CEPSA's Consolidated financial statements, 2022), installed solar panels at every service station, and launched new sustainable chemical products, NextLAB and NextPhen.

## 5. Conclusions

After analysing CEPSA's environmental and social responsibility performance between 2018 and 2022, the authors found that CEPSA has established a relatively systematic sustainability management system and an energy transition plan over five years. Regarding environmental performance, GHG emissions from CEPSA's two main energy parks decreased by 14 per cent. CEPSA has also made substantial progress in water resource management, renewable energy use and management, biodiversity and ecosystem maintenance, and waste management, with significant results. In terms of social responsibility performance, CEPSA focuses on disadvantaged groups, assessing employee performance and providing career development opportunities through an integrated talent management and evaluation model, and setting up a special committee to safeguard the lives and health of employees during COVID-19. Meanwhile, CEPSA built a good relationship with communities through the Sumamos Energias programme, developed low-carbon products through the 'Positive Movement' program, and formulated an ESG competence evaluation system for suppliers to help them improve their sustainable development capability.

Analysing CEPSA's financial performance from 2018 to 2022 using DuPont analysis reveals that CEPSA's return on net assets and equity have improved over the five years of the ESG transition. However, the financing structure has also changed, with debt financing taking up a much larger share than equity financing. The main factors contributing to this set of changes are income, debt, and investment [50]. Revenues fell sharply in 2020 due to COVID-19. However, the 'positive motion' program effectively boosted CEPSA's revenue. The liability increase is mainly due to the acquisition of petrol stations and new equipment. Following implementing an integrated management system for ESG performance, CEPSA found that existing $CO_2$ emissions exceeded $CO_2$ emission quotas, resulting in a shortfall in the environmental provision.

After analysing the financial performance metrics of CEPSA and Shell over multiple years, it is evident that CEPSA consistently exhibits a higher ROE than Shell, which indicates that CEPSA utilises its shareholder equity more efficiently to generate profits.

Both companies have shown a positive ROA over the years, but CEPSA tends to have a higher ROA, which reflects its proficiency in generating profits relative to its total assets. On the other hand, Shell consistently maintains a higher earnings margin than CEPSA, which indicates Shell's ability to produce higher profits per unit of sales. Despite fluctuations, Shell consistently demonstrates a higher net profit margin on sales compared to CEPSA, suggesting better profitability in terms of sales revenue. CEPSA exhibits a higher total asset turnover ratio than Shell, which indicates CEPSA's capability to generate more sales revenue relative to its total assets. CEPSA and Shell have low debt-to-asset ratios, indicating sound financial health and prudent debt management practices.

In conclusion, while CEPSA demonstrates strong performances in certain financial metrics like ROE and ROA, Shell maintains higher profitability ratios such as earnings margin and net profit margin on sales. Both companies exhibit prudent debt management practices reflected in their low debt-to-asset ratios. These findings suggest different areas of strength and opportunities for improvement for each company, contributing to a nuanced understanding of their financial performance.

In contrast, potential environmental remediation costs in the refinery area rose. The rapid asset growth is related to CEPSA's investment in energy development and rehabilitation projects. However, this paper also has limitations. Firstly, it discusses only the changes in CEPSA's ESG and financial performance, and the selected company has a better performance in the industry, so it needs to be more generalisable. Although CEPSA's ESG management concept gradually established only in 2020, the firm had a good foundation of sustainable development before, so the transition was relatively easy. Secondly, the information collection may need to be more comprehensive; the financial and non-financial information in this paper mainly comes from CEPSA's published annual integrated management report and consolidated financial statement, and they may be hiding some unfavourable information. The researchers conducted a Dupont analysis to evaluate energy companies' environmental and social performances to obtain answers for RQ1. The analysis showed that these indicators provide a comprehensive view of an energy company's ESG performance, which enables stakeholders to assess a company's sustainability practices and contributions to environmental and social well-being. Energy companies must track, report, and continuously improve these indicators to enhance their ESG performance and maintain stakeholder trust and confidence. For example, in 2022, CEPSA established the Sumamos Energias initiative to maximise the integration of its renewable energy installations into their surroundings and generate opportunities for the economic growth of local communities. CEPSA also sends experts during exploration and production operations to assist community members in increasing their project participation and focus on specific stakeholders. The research outcomes answer RQ2, stating that energy companies want to transform themselves. At the same time, controlling costs should prioritise a range of ESG activities that enhance sustainability and contribute to operational efficiency and cost savings. By implementing energy-efficient measures across operations, such as equipment upgrades, process optimisation, and energy management systems, energy consumption and operating costs can be lowered. For instance, CEPSA's new strategy puts customers at the centre of decarbonisation by offering a wide range of clean products, including wholesale B2B, aviation, lubricants, bitumen, LPG, natural gas, and electricity. The company also creates a value chain around biofuel, hydrogen, and renewable energy (solar and wind) businesses to help customers build decarbonisation solutions (CEPSA's integrated management report, 2022).

Moreover, between 2021 and 2022, CEPSA entered into strategic partnerships with Iberia Express and Seville Airport to develop and mass produce Sustainable Aviation Biofuels (SAFs). The partnership aims to advance the decarbonisation of the aviation industry by using waste, used oils, and second-generation biofuels. It has been discovered that sustainable development strategies can lead to improved financial performances for energy companies. This can be achieved through various mechanisms such as cost savings, revenue growth, risk mitigation, and better access to capital. By aligning business objectives

with sustainability goals and adopting innovative solutions, energy companies can achieve positive financial outcomes while advancing environmental and social objectives, which answers RQ3.

### 5.1. Limitations and Future Recommendations

This study is not free from limitations. The study focused on CEPSA, which may have unique characteristics compared to other companies in the industry, limiting the generalisability of findings to the broader energy sector. Therefore, it is recommended that future researchers conduct longitudinal studies to track the evolution of ESG and financial performances over an extended period, providing insights into long-term sustainability strategies and outcomes. Moreover, the current study relied on data primarily sourced from CEPSA's published reports, which may capture only some relevant information or could be subject to reporting biases, potentially limiting the comprehensiveness and accuracy of the analysis. Thus, complementing the scope of the current study with quantitative analyses and/or qualitative research methods can help future researchers gain a deeper understanding of the underlying factors driving changes in ESG and financial performances. This study highlights discrepancies in $CO_2$ emissions and potential environmental remediation costs, suggesting potential financial risks that warrant further investigation and analysis. Furthermore, due to time and word limitations, this study needs a further analysis of the corporate governance performance in the ESG performance section, and thus, the analysis may need to be completed. It is recommended that future researchers perform a comparative analysis of the sample used in the current study, which evaluates the company's financial performance concerning its industry peers. This comparative approach will provide additional insights into the effectiveness of the DuPont analysis framework and enhance the validity of our findings.

### 5.2. Implications

Investors can use the findings of this study to evaluate the long-term sustainability and financial performance of CEPSA, taking into account environmental, social, and governance (ESG) considerations while making investment decisions. CEPSA's commitment to ESG initiatives indicates its resilience to environmental and social risks, which can potentially decrease investment risk and enhance long-term returns for investors. Policymakers can also benefit from the study by using the insights gained to develop regulations and policies that promote sustainable business practices, improve transparency and accountability in corporate reporting, and incentivise companies to adopt sustainable development strategies.

Therefore, policymakers can consider supporting initiatives that encourage companies like CEPSA to reduce their environmental footprint further, such as renewable energy incentives and carbon pricing mechanisms, to facilitate the transition to a low-carbon economy. By analysing specific ESG initiatives implemented by CEPSA and their impacts on financial performance, the company can use the findings to inform strategic decision-making and resource allocation within the organisation. This will help CEPSA evaluate the effectiveness of its sustainability management system and energy transition plan, identify areas of success and improvement, and enhance stakeholder engagement efforts.

Additionally, this study can help CEPSA communicate its sustainability achievements more effectively, strengthen its reputation as a responsible corporate citizen, and identify potential financial risks, such as $CO_2$ emissions exceeding quotas and rising environmental remediation costs. In conclusion, this study underscores the importance of integrating ESG considerations into regulatory frameworks to encourage companies to adopt sustainable development strategies, improve their environmental and social performance, and manage potential financial risks more effectively.

**Author Contributions:** Conceptualisation, Y.H. and A.H.; methodology, Y.H. and A.H.; software, Y.H. and A.H.; validation, Y.H. and A.H.; formal analysis, Y.H., S.A. and A.H.; investigation, Y.H., S.A. and A.H.; resources, Y.H. and A.H.; data curation, Y.H., S.A. and A.H.; writing—original draft

preparation, S.A., Y.H. and A.H.; writing—review and editing, S.A., Y.H. and A.H.; visualisation, Y.H. and A.H.; supervision, S.A. and A.H.; project administration, S.A. and A.H. All authors have read and agreed to the published version of the manuscript.

**Funding:** This research received no external funding.

**Data Availability Statement:** The original contributions presented in this study are included in the article; further inquiries can be directed to the corresponding author.

**Conflicts of Interest:** The authors declare no conflicts of interest.

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
