# Peer review of "Examining the Interplay between CEPSA’s ESG Performance and Financial Performance: An Overview of the Energy Sector Transformation"

_sustainability, doi:10.3390/su16072772_

Round 1

Reviewer 1 Report

Comments and Suggestions for Authors

Edits:

Line 57:  capitalize Large-sample

Line 134: Lyndenberg

Line 130-131: there are often different evaluation methods instead of different scholars often have different…

Line 144: MSCI

Line 168: Lins

Line 323-324: delete “Because of time and word limitations,”

TABLE 2: Reformat. Either landscape the page or use vertical writing in both Group and Section

Line 352: scope 1 (add space)

Figure 2: remove in figure percentages and lines (think about bar chart)

Figure 3: remove in figure numbers (messy)

Line 528: remove space, 30%

Figure 4: remove in figure numbers

Remove figure 5 not enough data (describe in text)

Remove figure 6 not enough data (describe in text)

Figure 7: Y-axis only one decimal (1.0, 0.0, …)

Table 5: Clean up by creating a note of ROE(1) = ROA(t)*EM(t), ROE(2)=ROA(t+1)*EM(t), ROE(3)=ROA(t+1)*EM(t+1), Impact of ROA = ROE(2)-ROE(1), Impact of EM = ROE(3)-ROE(2)

                Table then becomes Years, ROE(1), ROE(2), ROE(3), and Impact ROA, Impact EM

Line 747: noting not noticing

Table 6: redo along the lines of Table 5 suggestions

Line 792: Change

The authors offer 3 research questions that they ultimately do not answer (lines 272-279). Most of the analysis is descriptive. Any quantitative analysis comes down to tables 5 and 6 which are ill presented and also only a simple rearrangement of DuPont identity analysis. 

Comments on the Quality of English Language

See edits above.

Author Response

Reviewer 1

We sincerely thank reviewer 1 for their diligent efforts in reviewing our manuscript and providing valuable and constructive feedback that helped us enhance its quality. We have carefully revised the manuscript to address the concerns and suggestions raised during the review process. Below, we have provided our responses to the reviewers' comments and questions. The reviewers' comments and questions are in “Black”, and our responses are in “Red”.

Line 57:  capitalize Large sample.

Comment: Thank you for your suggestion. We have revised Line 57 to capitalise 'Large sample' as requested.

Line 134: Lyndenberg

Comment: Thank you for your suggestion. We have made the required changes.

Line 130-131: there are often different evaluation methods instead of different scholars often have different.

Comment: Thank you for your suggestion. We have made the required changes.

Line 144: MSCI

Comment: Thank you for your suggestion. We have made the required changes.

Line 168: Lins

Comment: Thank you for your suggestion. We have made the required changes.

Line 323-324: delete “Because of time and word limitations,”

Comment: Thank you for your suggestion. We have made the required changes.

TABLE 2: Reformat. Either landscape the page or use vertical writing in both Group and Section

Comment: Thank you for your suggestion. We have made the required changes.

Line 352: scope 1 (add space)

Comment: Thank you for your suggestion. We have made the required changes.

Figure 2: remove in figure percentages and lines (think about bar chart)

Comment: Thank you for your suggestion. We have made the required changes.

Figure 3: remove in figure numbers (messy)

Comment: Thank you for your suggestion. We have made the required changes.

Line 528: remove space, 30%

Comment: Thank you for your suggestion. We have made the required changes.

Figure 4: remove in figure numbers

Comment: Thank you for your suggestion. We have made the required changes.

Remove figure 5 not enough data (describe in text)

Comment: Thank you for your suggestion. We have made the required changes.

Remove figure 6 not enough data (describe in text)

Comment: Thank you for your suggestion. We have made the required changes.

Figure 7: Y-axis only one decimal (1.0, 0.0, …)

Comment: Thank you for your suggestion. We have made the required changes.

Table 5: Clean up by creating a note of ROE(1) = ROA(t)*EM(t), ROE(2)=ROA(t+1)*EM(t), ROE(3)=ROA(t+1)*EM(t+1), Impact of ROA = ROE(2)-ROE(1), Impact of EM = ROE(3)-ROE(2)

                Table then becomes Years, ROE(1), ROE(2), ROE(3), and Impact ROA, Impact EM

Comment: Thank you for your suggestion. We have made the required changes.

Line 747: noting not noticing

Comment: Thank you for your suggestion. We have made the required changes.

Table 6: redo along the lines of Table 5 suggestions

Comment: Thank you for your suggestion. We have made the required changes.

Line 792: Change

Comment: Thank you for your suggestion. We have made the required changes.

The authors offer 3 research questions that they ultimately do not answer (lines 272-279). Most of the analysis is descriptive. Any quantitative analysis comes down to tables 5 and 6 which are ill presented and also only a simple rearrangement of DuPont identity analysis. 

Comment: We appreciate your thoughtful comments on the need to ensure that all posed research questions are fully explored and answered, particularly regarding the nuanced ways in which ESG initiatives contribute to financial metrics. We have added following explanation in the manuscript (conclusion):

Line 899-930: “The researchers conducted a Dupont analysis to evaluate energy companies' environmental and social performance to obtain answers for RQ1. The analysis showed that these indicators provide a comprehensive view of an energy company's ESG performance, which enables stakeholders to assess its sustainability practices and contributions to environmental and social well-being. Energy companies must track, report, and continuously improve these indicators to enhance their ESG performance and maintain stakeholder trust and confidence. For example, in 2022, CEPSA established the Sumamos Energias initiative to maximize the integration of its renewable energy installations into their surroundings and generate opportunities for the economic growth of local communities. CEPSA also sends experts during exploration and production operations to assist community members in increasing their project participation and focusing on specific stakeholders. The research outcomes answer RQ2, stating that energy companies want to transform themselves. At the same time, controlling costs should prioritize a range of ESG activities that enhance sustainability and contribute to operational efficiency and cost savings. By implementing energy-efficient measures across operations, such as equipment upgrades, process optimization, and energy management systems, energy consumption and operating costs can be lowered. For instance, CEPSA's new strategy puts customers at the centre of decarbonization by offering a wide range of clean products, including wholesale B2B, aviation, lubricants, bitumen, LPG, natural gas, and electricity. The company also creates a value chain around biofuels, hydrogen, and renewable energy (solar and wind) businesses to help customers build decarbonization solutions (CEPSA’s integrated management report, 2022).

Moreover, between 2021 and 2022, CEPSA has entered into strategic partnerships with Iberia Express and Seville Airport to develop and mass-produce Sustainable Aviation Biofuels (SAF). The partnership aims to advance the decarbonization of the aviation industry by using waste, used oils, and second-generation biofuels. It has been discovered that sustainable development strategies can lead to improved financial performance for energy companies. This can be achieved through various mechanisms such as cost savings, revenue growth, risk mitigation, and better access to capital. By aligning business objectives with sustainability goals and adopting innovative solutions, energy companies can achieve positive financial outcomes while advancing environmental and social objectives, which answers RQ3.”

Additionally, the primary aim of our study was to provide a comprehensive overview of the topic and lay the groundwork for future research in this area. Thus, we focus on descriptive analysis to reflect our intention to provide a thorough examination of the subject matter. As for the quantitative analysis, we acknowledge that tables 5 and 6 present a simplified rearrangement of the DuPont identity analysis. Therefore, we have added this as a recommendation for future researcher such as:

Line 946-952: “Furthermore, due to time and word limitations, this paper needs an analysis of corporate governance performance in the ESG performance section, and thus, the analysis may need to be completed. It is recommended that future researchers perform a comparative analysis of the sample used in the current study, which evaluates the company's financial performance concerning its industry peers. This comparative approach will provide additional insights into the effectiveness of the DuPont analysis framework and enhance the validity of our findings.”

Thank you. 

Reviewer 2 Report

Comments and Suggestions for Authors

Your paper aims to understand how CEPSA's ESG performance influenced its financial metrics over the last five years. It tries to investigate the effect of ESG management on financial performance indicators using DuPont analysis. It addresses the gap in literature regarding detailed case studies that link ESG initiatives directly to financial performance metrics. It offers a nuanced understanding of how specific ESG actions impact various financial performance indicators, enriching the discourse on sustainable development within the energy sector.

Providing more detail about the DuPont analysis method and its application could strengthen the academic rigor of the paper. While the use of DuPont analysis is appropriate, you should enhance the methodology by incorporating a comparative analysis with peer companies. This would provide a benchmark for understanding CEPSA's performance in the context of the industry standard. Additionally, considering control variables such as market conditions or regulatory changes could deepen the analysis.

The paper should address whether all posed research questions were fully explored and answered, particularly the nuanced ways in which ESG initiatives specifically contributed to financial metrics. I recommend that the authors elaborate extensively on the specific ESG initiatives that CEPSA has implemented. This should include a detailed description of each initiative, outlining its scope, objectives, and the strategies employed to achieve these goals. Furthermore, you should analyze the direct and indirect impacts of these ESG actions on CEPSA's financial performance, providing a comprehensive evaluation of how each initiative contributes to financial metrics such as profitability, revenue growth, cost savings, and risk mitigation. This detailed exposition will not only clarify the causal relationships between ESG initiatives and financial outcomes but also enrich the study's contribution to understanding the business value of ESG investments.

The references appear suitable and relevant. Still, incorporating more recent studies could strengthen the paper, especially those focusing on the latest trends in ESG reporting and its financial implications post-2022.

You must significantly enhance the visual presentation and interpretative clarity of the tables and figures utilized throughout the paper. This involves a meticulous redesign to ensure that each graphical element not only effectively communicates the underlying data but also does so in a manner that is immediately comprehensible to the reader. Attention should be given to the use of color, font sizes, labeling, details (percentages, etc,) and the inclusion of explanatory legends where necessary. You should move table 1 as an annex. Pay attention to figures 2,3,4.

You should add a more detailed discussion of the study's limitations and potential biases that could enhance the paper's academic credibility. Discuss any limitations related to data availability, selection, and potential biases that might affect the analysis! By acknowledging these aspects explicitly, you would provide readers with a clearer understanding of the context and constraints under which the study was conducted, thereby enhancing the reliability and validity of the research findings. At the same time, elaborating on specific areas for future research based on the study's findings could be beneficial for advancing the field.

Discuss the implications of your findings for various stakeholders, including investors, policy makers, and the company itself.

Author Response

Reviewer 2

We sincerely thank reviewer 2 for their diligent efforts in reviewing our manuscript and providing valuable and constructive feedback that helped us enhance its quality. We have carefully revised the manuscript to address the concerns and suggestions raised during the review process. Below, we have provided our responses to the reviewers' comments and questions. The reviewers' comments and questions are in “Black”, and our responses are in “Red”.

Your paper aims to understand how CEPSA's ESG performance influenced its financial metrics over the last five years. It tries to investigate the effect of ESG management on financial performance indicators using DuPont analysis. It addresses the gap in literature regarding detailed case studies that link ESG initiatives directly to financial performance metrics. It offers a nuanced understanding of how specific ESG actions impact various financial performance indicators, enriching the discourse on sustainable development within the energy sector.

Comment: We would like to express our gratitude to the reviewer for their constructive feedback.

Providing more detail about the DuPont analysis method and its application could strengthen the academic rigor of the paper. While the use of DuPont analysis is appropriate, you should enhance the methodology by incorporating a comparative analysis with peer companies. This would provide a benchmark for understanding CEPSA's performance in the context of the industry standard. Additionally, considering control variables such as market conditions or regulatory changes could deepen the analysis.

Comment: Thank you for your valuable feedback. We appreciate your suggestion to provide more detail about the DuPont analysis method and its application in our study. In response to this suggestion, we have made the following addition to the text:

Line 307-320: "The researcher has incorporated DuPont analysis to conduct the current study, a financial analysis framework that breaks down the return on equity (ROE) into its parts. The DuPont Corporation developed it in the early 20th century to assess the drivers of profitability and efficiency in a company's operations. The reason behind selecting this analysis is its ability to provide insights into the drivers of profitability and financial performance within a company.

In the current study, by breaking down ROE into parts, DuPont analysis has helped the researchers to identify the specific areas contributing to a company's overall profitability and efficiency. Moreover, it helped the researchers assess the impact of various factors, such as Environmental performance, water management, other renewable energy, environmental management, employees and collaboration on a company's financial performance. Thus, DuPont analysis seems the most suited for the current study as it served as a valuable tool for understanding the drivers of profitability and environmental performance of sample companies."

Furthermore, we acknowledge the importance of incorporating a comparative analysis with peer companies to strengthen the methodology. Thus, we have included this as a recommendation for future researchers such as:

Line 946-950: “It is recommended that future researchers perform a comparative analysis of the sample used in the current study, which evaluates the company's financial performance concerning its industry peers. This comparative approach will provide additional insights into the effectiveness of the DuPont analysis framework and enhance the validity of our findings.”

The paper should address whether all posed research questions were fully explored and answered, particularly the nuanced ways in which ESG initiatives specifically contributed to financial metrics. I recommend that the authors elaborate extensively on the specific ESG initiatives that CEPSA has implemented. This should include a detailed description of each initiative, outlining its scope, objectives, and the strategies employed to achieve these goals. Furthermore, you should analyze the direct and indirect impacts of these ESG actions on CEPSA's financial performance, providing a comprehensive evaluation of how each initiative contributes to financial metrics such as profitability, revenue growth, cost savings, and risk mitigation. This detailed exposition will not only clarify the causal relationships between ESG initiatives and financial outcomes but also enrich the study's contribution to understanding the business value of ESG investments.

Comment: We appreciate your thoughtful comments on the need to ensure that all posed research questions are fully explored and answered, particularly regarding the nuanced ways in which ESG initiatives contribute to financial metrics. We have added following explanation in the manuscript:

Line 899-930: “The researchers conducted a Dupont analysis to evaluate energy companies' environmental and social performance to obtain answers for RQ1. The analysis showed that these indicators provide a comprehensive view of an energy company's ESG performance, which enables stakeholders to assess its sustainability practices and contributions to environmental and social well-being. Energy companies must track, report, and continuously improve these indicators to enhance their ESG performance and maintain stakeholder trust and confidence. For example, in 2022, CEPSA established the Sumamos Energias initiative to maximize the integration of its renewable energy installations into their surroundings and generate opportunities for the economic growth of local communities. CEPSA also sends experts during exploration and production operations to assist community members in increasing their project participation and focusing on specific stakeholders. The research outcomes answer RQ2, stating that energy companies want to transform themselves. At the same time, controlling costs should prioritize a range of ESG activities that enhance sustainability and contribute to operational efficiency and cost savings. By implementing energy-efficient measures across operations, such as equipment upgrades, process optimization, and energy management systems, energy consumption and operating costs can be lowered. For instance, CEPSA's new strategy puts customers at the centre of decarbonization by offering a wide range of clean products, including wholesale B2B, aviation, lubricants, bitumen, LPG, natural gas, and electricity. The company also creates a value chain around biofuels, hydrogen, and renewable energy (solar and wind) businesses to help customers build decarbonization solutions (CEPSA’s integrated management report, 2022).

Moreover, between 2021 and 2022, CEPSA has entered into strategic partnerships with Iberia Express and Seville Airport to develop and mass-produce Sustainable Aviation Biofuels (SAF). The partnership aims to advance the decarbonization of the aviation industry by using waste, used oils, and second-generation biofuels. It has been discovered that sustainable development strategies can lead to improved financial performance for energy companies. This can be achieved through various mechanisms such as cost savings, revenue growth, risk mitigation, and better access to capital. By aligning business objectives with sustainability goals and adopting innovative solutions, energy companies can achieve positive financial outcomes while advancing environmental and social objectives, which answers RQ3.”

The references appear suitable and relevant. Still, incorporating more recent studies could strengthen the paper, especially those focusing on the latest trends in ESG reporting and its financial implications post-2022. You must significantly enhance the visual presentation and interpretative clarity of the tables and figures utilized throughout the paper. This involves a meticulous redesign to ensure that each graphical element not only effectively communicates the underlying data but also does so in a manner that is immediately comprehensible to the reader. Attention should be given to the use of color, font sizes, labeling, details (percentages, etc,) and the inclusion of explanatory legends where necessary. You should move table 1 as an annex. Pay attention to figures 2,3,4.

Comment: Thank you for your suggestions. We have made required changes to the suggested tables and figures. Additionally, we have included 7 recent references.

You should add a more detailed discussion of the study's limitations and potential biases that could enhance the paper's academic credibility. Discuss any limitations related to data availability, selection, and potential biases that might affect the analysis! By acknowledging these aspects explicitly, you would provide readers with a clearer understanding of the context and constraints under which the study was conducted, thereby enhancing the reliability and validity of the research findings. At the same time, elaborating on specific areas for future research based on the study's findings could be beneficial for advancing the field.

Comment: Thank you for your valuable comment. We have added some limitations and recommendations for future researchers under a new heading i.e, 5.1. Such as:

Line 932-952: 5.1. Limitations and Future RecommendationsThis study is not free from limitations. The study focuses on CEPSA, which may have unique characteristics compared to other companies in the industry, limiting the generalizability of findings to the broader energy sector. Therefore, it is recommended that future researchers conduct longitudinal studies to track the evolution of ESG and financial performance over an extended period, providing insights into long-term sustainability strategies and outcomes. Moreover, the current study relies on data primarily sourced from CEPSA's published reports, which may only capture some relevant information or could be subject to reporting biases, potentially limiting the comprehensiveness and accuracy of the analysis. Thus, complementing the scope of the current study with quantitative analyses and/or qualitative research methods can help future researchers gain a deeper understanding of the underlying factors driving changes in ESG and financial performance. The study highlights discrepancies in CO2 emissions and potential environmental remediation costs, suggesting potential financial risks that warrant further investigation and analysis

Discuss the implications of your findings for various stakeholders, including investors, policy makers, and the company itself.

Comment: Thank you for your valuable comment. We have added some implications under a new heading i.e, 5.2.

Line 954-979: 5.2. Implications

Investors can use the findings of this study to evaluate the long-term sustainability and financial performance of CEPSA, taking into account environmental, social, and governance (ESG) considerations while making investment decisions. CEPSA's commitment to ESG initiatives indicates its resilience to environmental and social risks, which can potentially decrease investment risk and enhance long-term returns for investors. Policymakers can also benefit from the study by using the insights gained to develop regulations and policies that promote sustainable business practices, improve transparency and accountability in corporate reporting, and incentivize companies to adopt sustainable development strategies.

Therefore, policymakers can consider supporting initiatives that encourage companies like CEPSA to reduce their environmental footprint further, such as renewable energy incentives and carbon pricing mechanisms, to facilitate the transition to a low-carbon economy. By analyzing specific ESG initiatives implemented by CEPSA and their impacts on financial performance, the company can use the findings to inform strategic decision-making and resource allocation within the organization. This will help CEPSA evaluate the effectiveness of its sustainability management system and energy transition plan, identify areas of success and improvement, and enhance stakeholder engagement efforts.

Additionally, the study can help CEPSA communicate its sustainability achievements more effectively, strengthen its reputation as a responsible corporate citizen, and identify potential financial risks such as CO2 emissions exceeding quotas and rising environmental remediation costs. In conclusion, this study underscores the importance of integrating ESG considerations into regulatory frameworks to encourage companies to adopt sustainable development strategies, improve their environmental and social performance, and manage potential financial risks more effectively.

Thank you. 

Reviewer 3 Report

Comments and Suggestions for Authors

The paper investigates the financial performance of the Spanish energy company CEPSA by evaluating changes in financial indicators for the period 2018-2022 with the help of DuPont analysis.

In the literature, there are quite a few articles that individually analyzed each large company on different research topics.

Considering that it is a widely debated topic, the literature could be updated with articles published in 2023, below are some relevant articles.

Ates, S. (2023). The credibility of corporate social responsibility reports: evidence from the energy sector in emerging markets. Social Responsibility Journal19(4), 756-773.

Santos, M. R., Rolo, A., Matos, D., & Carvalho, L. (2023). The Circular Economy in Corporate Reporting: Text Mining of Energy Companies’ Management Reports. Energies16(15), 5791.

It would be good if you managed to make the work more attractive for readers, both as a graphic presentation of the figures and tables, as well as content by making some comparisons with other profile companies in Europe.

The results obtained were discussed, but in the discussion section I would have liked to read a broader debate on the results obtained by the companies from the other countries, similarities, differences, on how the results can be generalized. Likewise, managerial implications can be developed.

The limits of the theoretical study and future research directions should also be added.

Author Response

Reviewer 3

We sincerely thank reviewer 3 for their diligent efforts in reviewing our manuscript and providing valuable and constructive feedback that helped us enhance its quality. We have carefully revised the manuscript to address the concerns and suggestions raised during the review process. Below, we have provided our responses to the reviewers' comments and questions. The reviewers' comments and questions are in “Black”, and our responses are in “Red”.

The paper investigates the financial performance of the Spanish energy company CEPSA by evaluating changes in financial indicators for the period 2018-2022 with the help of DuPont analysis. In the literature, there are quite a few articles that individually analyzed each large company on different research topics.Considering that it is a widely debated topic, the literature could be updated with articles published in 2023, below are some relevant articles.

Ates, S. (2023). The credibility of corporate social responsibility reports: evidence from the energy sector in emerging markets. Social Responsibility Journal19(4), 756-773.

Santos, M. R., Rolo, A., Matos, D., & Carvalho, L. (2023). The Circular Economy in Corporate Reporting: Text Mining of Energy Companies’ Management Reports. Energies16(15), 5791.

Comment: Thank you for your suggestion. We have added five more recent references along with these references to support important statements.

It would be good if you managed to make the work more attractive for readers, both as a graphic presentation of the figures and tables, as well as content by making some comparisons with other profile companies in Europe.

Comment: Thank you for your suggestions. We have made required changes to the suggested tables and figures.

The results obtained were discussed, but in the discussion section I would have liked to read a broader debate on the results obtained by the companies from the other countries, similarities, differences, on how the results can be generalized. Likewise, managerial implications can be developed.

Comment: Thank you for your valuable comment. We have added some implications under a new heading i.e, 5.2.

Line 954-979: 5.2. Implications

Investors can use the findings of this study to evaluate the long-term sustainability and financial performance of CEPSA, taking into account environmental, social, and governance (ESG) considerations while making investment decisions. CEPSA's commitment to ESG initiatives indicates its resilience to environmental and social risks, which can potentially decrease investment risk and enhance long-term returns for investors. Policymakers can also benefit from the study by using the insights gained to develop regulations and policies that promote sustainable business practices, improve transparency and accountability in corporate reporting, and incentivize companies to adopt sustainable development strategies.

Therefore, policymakers can consider supporting initiatives that encourage companies like CEPSA to reduce their environmental footprint further, such as renewable energy incentives and carbon pricing mechanisms, to facilitate the transition to a low-carbon economy. By analyzing specific ESG initiatives implemented by CEPSA and their impacts on financial performance, the company can use the findings to inform strategic decision-making and resource allocation within the organization. This will help CEPSA evaluate the effectiveness of its sustainability management system and energy transition plan, identify areas of success and improvement, and enhance stakeholder engagement efforts.

Additionally, the study can help CEPSA communicate its sustainability achievements more effectively, strengthen its reputation as a responsible corporate citizen, and identify potential financial risks such as CO2 emissions exceeding quotas and rising environmental remediation costs. In conclusion, this study underscores the importance of integrating ESG considerations into regulatory frameworks to encourage companies to adopt sustainable development strategies, improve their environmental and social performance, and manage potential financial risks more effectively.

The limits of the theoretical study and future research directions should also be added

Comment: Thank you for your valuable comment. We have added some limitations and recommendations for future researchers under a new heading i.e, 5.1. Such as:

Line 932-952: 5.1. Limitations and Future Recommendations This study is not free from limitations. The study focuses on CEPSA, which may have unique characteristics compared to other companies in the industry, limiting the generalizability of findings to the broader energy sector. Therefore, it is recommended that future researchers conduct longitudinal studies to track the evolution of ESG and financial performance over an extended period, providing insights into long-term sustainability strategies and outcomes. Moreover, the current study relies on data primarily sourced from CEPSA's published reports, which may only capture some relevant information or could be subject to reporting biases, potentially limiting the comprehensiveness and accuracy of the analysis. Thus, complementing the scope of the current study with quantitative analyses and/or qualitative research methods can help future researchers gain a deeper understanding of the underlying factors driving changes in ESG and financial performance. The study highlights discrepancies in CO2 emissions and potential environmental remediation costs, suggesting potential financial risks that warrant further investigation and analysis.

Thank you. 

Round 2

Reviewer 1 Report

Comments and Suggestions for Authors

The authors have dutifully corrected their errors to the manuscript, but the level of the research has not been upgraded. No real progress has been made in the research only in the writing and presentation.

Author Response

Dear Reviewer, 

We appreciate your thorough review and constructive criticism.

We have carefully considered the suggestions provided, and in response, we have incorporated additional analysis into the manuscript, specifically addressing the comparative financial performance of the companies. You will find the detailed analysis in section 4.3.3 of the revised version.

We believe these additions significantly strengthen the paper and address the concerns raised by the reviewer. 

Thank you.

Reviewer 3 Report

Comments and Suggestions for Authors

The paper is improved, but  I did not find references to how to generalize the results compared to other profile companies in Spain (e.g. Repsol) or other European countries (BP, Shell, Total, OMV, etc.)

The information on the link below may also be useful.

https://www.knowesg.com/company-esg-ratings?company-esg-ratings%5Bmenu%5D%5Bsector%5D=Energy

Author Response

Dear Reviewer, 

We appreciate your thorough review and constructive criticism.

We have carefully considered the suggestions provided, and in response, we have incorporated additional analysis into the manuscript, specifically addressing the comparative financial performance of the companies. You will find the detailed analysis in section 4.3.3 of the revised version.

Additionally, we have included a comparison analysis in the conclusion, i.e., Line 951-Line 958.

We believe these additions significantly strengthen the paper and address the concerns raised by the reviewer. 

Thank you.

Round 3

Reviewer 1 Report

Comments and Suggestions for Authors

Very good progress.